



# The Northeast Greenland shelf as a late-summer $CO_2$ source to the atmosphere

Esdoorn Willcox[1], Marcos Lemes[1], Thomas Juul-Pedersen[3], Mikael Kristian Sejr[4], Johnna Michelle Holding[4], and Søren Rysgaard[2]

[1]Centre for Earth and Observation Science, University of Manitoba, Manitoba, Canada
[2]Arctic Research Centre, Aarhus University, Aarhus, Denmark
[3]Pinngortitaleriffik, Greenland Institute of Natural Resources, Kivioq 2, PO Box 570, 3900 Nuuk, Greenland
[4]Institut for ecoscience, Aarhus University, Aarhus, Denmark

**Correspondence:** Esdoorn Willcox (willcoxe@myumanitoba.ca)

**Abstract.** The Northeast Greenland shelf carbon system is largely undescribed with the exception of the region associated with the Northeast Water Polynya. We describe the carbon system and the dominant processes affecting it in the region between 24 August and 25 September 2017. During this period the shelf was largely sea ice free and although the north shelf was a carbon dioxide sink, the rest of the shelf and slope acted as both source and sink. This is in contrast to the common perception for this

Arctic outflow shelf region as a $CO_2$ sink during the ice-free season. In the southern end of our sampling area, and particularly along the slope, low values of TA can lead to the shelf being a strong carbon dioxide source to the atmosphere. We hypothesize on the possible causes for this low TA.

## 1 Introduction

Since the 1990s, the Northeast Greenland shelf has been determined to be an annual net sink for atmospheric carbon dioxide

($CO_2$). This determination is based primarily on a few observations (Tan et al., 1983; Yager et al., 1995; Jeansson et al., 2008; Sejr et al., 2011) and subsequent modeling efforts (Takahashi et al., 2014). Yager et al. (1995) used their data to develop the 'seasonal rectification hypothesis' which describes the region as a net annual $CO_2$ sink. Their hypothesis is based on the concept that large primary productivity and sea ice melt cause the region to function as a strong sink of atmospheric carbon dioxide in spring and early summer. During autumn, prior to the region becoming net heterotrophic, the region is covered in sea

ice which inhibits/limits the exchange of gases, including $CO_2$, with the atmosphere. Since the uptake in spring and summer is larger than the release during autumn and winter, there is a net annual uptake of $CO_2$. More recently, the timing of the onset of seasonal sea ice cover in the Arctic Ocean is increasingly delayed and therefore it is unclear whether the seasonal rectification hypothesis still applies to the Northeast Greenland shelf.

The Arctic Ocean is losing sea ice rapidly under the influence of Arctic amplification, i.e. the proportionally enhanced

increase in Arctic atmospheric temperatures compared to those at lower latitudes (Serreze and Barry, 2011). Atlantification of the Arctic ocean, the transport of warmer water to higher latitudes, is co-occurring with atmospheric warming. Increased temperatures in both ocean and atmosphere contribute to sea ice melt which, in the absence of energy available for mixing,





enhances upper layer stratification. Increases in river runoff also add freshwater to the Arctic Ocean and control seasonal stratification in the long term (Farmer et al., 2021).

Surface layer stratification can act as a double edged sword with respect to the carbon equilibrium between the surface mixed layer and the atmosphere depending on the carbon chemistry of this layer. The chemistry is determined by the composition of advected water with local proccesses such as precipitation and sea ice melt and freeze processes superimposed. After initial losses of solutes during initial frazil sea ice formation, temperatures in the new congelation ice layer at the water surface are sufficiently cold for the mineral precipiate ikaite ($CaCO_3 \cdot 6H_2O$) to form. The crystals of ikaite formed are thought to be

preferentially retained in the sea ice while the carbon continues to be lost through brine drainage (Rysgaard et al., 2013, 2014). When this layer melts it could thus release this calcium carbonate as excess alkalinity, compared to the initial seawater the ice was formed from, to the water column directly below (Geilfus et al., 2016) and enhance $CO_2$ uptake. The enhancement of alkalinity may be further enhanced by the presence of snow at the sea ice surface in which this process seems to be enhanced (Nedashkovsky et al., 2009; Søgaard et al., 2019). Usually this effect is considered to be small when considered mixed into

winter mixed layer depths of 70 m (Peralta-Ferriz and Woodgate, 2015). In a highly stratified environment with summer mixed layer depths of 10 to 30 m only, like on the Northeast Greenland shelf in late summer, a meltwater lens at the surface might promote the ocean surface absorption of $CO_2$.

Simultaneously, sea ice melt and other meteoric freshwater inputs that are low in nutrients can inhibit mixing by increasing the amount of energy required for mixing, though this effect may not be as large on the Northeast Greenland shelf as a result of

the shoaling of the Atlantic Water layer (Gjelstrup et al., 2022). Once nutrients have been used up after the initial spring bloom, this stratification has the potential to inhibit primary productivity in the surface layer and the associated carbon drawdown from the atmosphere. Carbon will continue to be formed and exported below the mixed layer as part of the productivity taking place in the deep chlorophyll maximum but due to being separated from the atmosphere will not be able to utilize atmospheric carbon, and rather use remineralised forms. During the polar night, light-driven autotrophy is not possible and the Arctic is

considered to be net heterotrophic (Yager et al., 1995). If sea ice forms a cap across the ocean, mostly limiting heterotrophy-associated outgassing of $CO_2$, outgasssing is not sufficient to balance the large productivity seen during spring (and potentially summer and autumn) bloom(s). The later sea ice is formed, the more likely it is for heterotrophic outgassing of $CO_2$ to start balancing out autotrophic uptake, leading to a lower annual net uptake of $CO_2$ overall (Yager et al., 1995).

This paper focuses on describing observations made of alkalinity and dissolved inorganic carbon during late summer and

early autumn of 2017. Since these observations were made during a period of low sea ice cover, they provide a possible insight into the response in terms of $CO_2$ exchange on the Northeast Greenland shelf to a changing climate.

## 2    Materials & methods

### 2.1    Cruise location and sample analysis

Data for this study was collected during two cruises (DANA2017 and NEGREEN 2017) (Figure 1). CTD stations were placed

into groups based on their temperature and salinity characteristics, resulting in 5 groups that matched different geographical



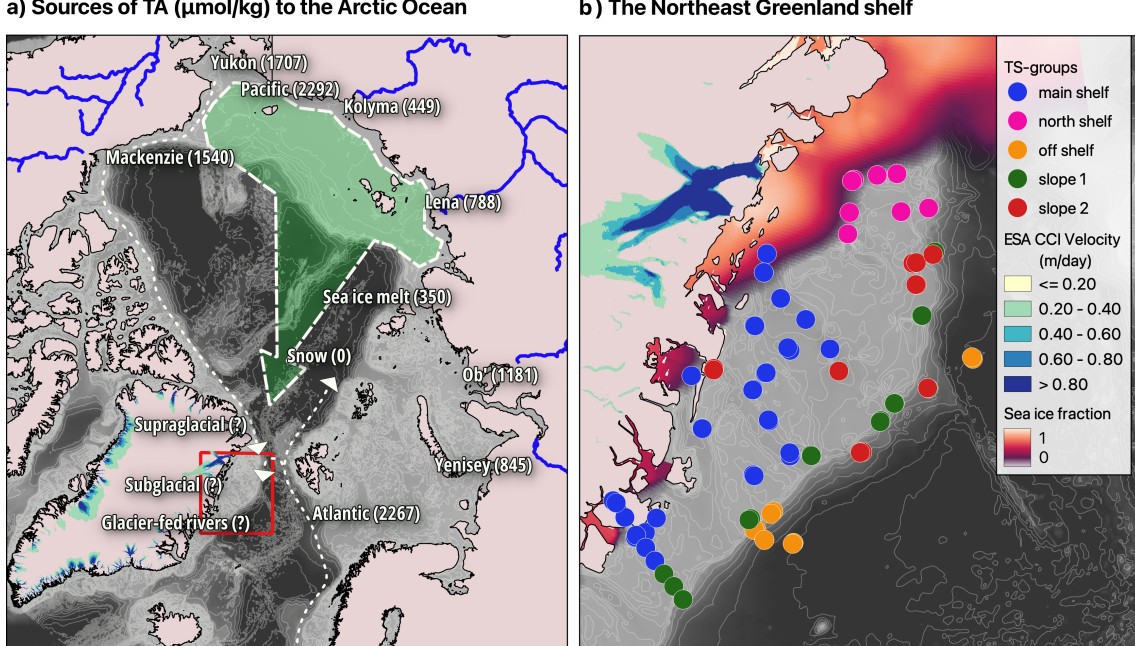

**Figure 1.** a: Known sources of total alkalinity to the Arctic Ocean highlighting the source regions of the Transpolar drift (green area with dashed white outline) and the location of the study area (red rectangle). Sources to the Arctic Ocean include Arctic rivers with variable catchment geology, sea ice and snow melt, and the Pacific Water coming in through the Bering Strait. River values were taken from Cooper et al. (2008) and Pacific water from Anderson et al. (2013). Locally, there is an unknown contribution of both sub- and supraglacial sources as well as glacier-fed rivers. b: CTD Stations and their associated TS-group on the Northeast Greenland shelf per Willcox et al. (2023) with the fraction of sea ice concentration on the 13th of September 2017 in red. Bathymetry was sourced from IBCAO (Jakobsson et al., 2020), sea ice extent from OSTIA (Good et al., 2020), and ice velocity from QGreenland v2 (Moon et al., 2022)

areas as shown in (Figure 1 b). These groups will be referred to as TS-groups since they are based on their *TS* profiles for the remainder of the document. For more detail regarding the cruises, CTD station data processing, stable water isotope ($\delta^{18}$O), and total alkalinity analyses, please refer to our previous paper, Willcox et al. (2023).

To analyse DIC, seawater samples were transferred to gas-tight vials (12 mL Exetainer, Labco High Wycombe,UK), poisoned
with 12 µL solution of saturated $HgCl_2$, and stored in the dark at room temperature until analysis. DIC was measured on a DIC analyzer (Apollo SciTech, Newark, DE, USA) by acidification of a 0.75 mL subsample with 1 mL 10% $H_3PO_4$ (Sigma-Aldrich, Saint-Louis, MO, USA), and quantification of the released $CO_2$ with a nondispersive infrared $CO_2$ analyzer (LI-COR, LI-7000, Lincoln, NE, USA). Results were then converted from mmol $L^{-1}$ to mmol $kg^{-1}$ based on sample density, which was estimated from salinity and temperature. An accuracy of  2 mmol $kg^{-1}$ was determined for DIC from routine analysis of
certified reference material (A.G. Dickson, Scripps Institution of Oceanography, San Diego, CA, USA).





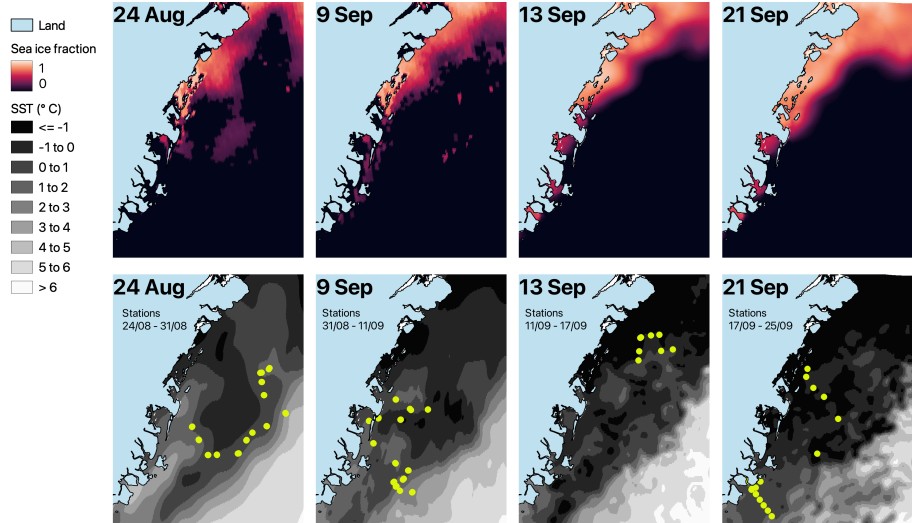

**Figure 2.** Surface conditions on the shelf through the period of data collection. Top row shows sea ice conditions, bottom row is sea surface temperature. Representative dates from left to right are the start date of collection (24 August), mid-collection (13 September) and near-end of data collection (21 September). Both figures are obtained from Meteorological Office UK (2019) (Good et al., 2020). The increased spatial patterning is due to cloud cover during the first period and reliance solely on lower resolution data

CTD data combined with the TA and DIC were used to calculate the pCO2 using the model program CO2SYS (van Heuven et al., 2011) using the dissociation constants of Mehrbach et al. (1973) refitted by Dickson and Millero (1987) and the hydrogen sulfite dissociation constant from Dickson (1990).

## 2.2 Mixed layer depth determination

To investigate the potential direction of gaseous transfer between the atmosphere and ocean it is important to estimate the depth of the mixed layer since this is the layer where such interactions take place. There are many different ways to determine mixed layer depth and ours is highly simplified. For each CTD station for which carbonate data was available, the mixed layer depth was assumed to be less than 70 m since that is the average expected winter mixed layer depth in the Eurasian basin where much of our surface water is thought to originate (Peralta-Ferriz and Woodgate, 2015). Above that depth we decided that convection is most likely to happen at depths shallower than the maximum Brunt-Vaiisala frequency squared ($N^2$) at that station. This is somewhat plausible considering not only the temperature and salinity profiles, but also that the dissolved oxygen maximum is directly below this depth and providing evidence that gases are trapped at depths below this as described previously (Willcox et al., 2023).





### 2.3 Meteoric and sea ice melt fractions

The determination of meteoric and sea ice melt fractions for this dataset in the Northeast Greenland shelf is far from straightforward. Not only are there multiple sources of meteoric freshwater with different contributions to the alkalinity and to the stable water isotopic composition ($\delta^{18}$O, $\delta^2$H) but also due to processes taking place on the shelf itself. Even if the primary source location for the freshwater found on the Northeast Greenland shelf is the Laptev Sea and the Lena river in late summer 2017 (Willcox et al., 2023), processes such as sulphate reduction and denitrification of organic matter have the potential to change

the alkalinity during cross-shelf transport (Middelburg et al., 2020) prior to advection into the Transpolar Drift The three main sources (end-members) are shown in (Table 1), where Atlantic water includes water in the Return Atlantic Current which is advected directly from the North-flowing West-Spitsbergen Current, as well as Arctic-sourced Eurasian Basin Atlantic Water (Figure 1). These have different associated end-member tracer values including for $\delta^{18}$O, alkalinity, and salinity. The Meteoric fraction similarly includes end-members with different, and potentially seasonally variable tracer end-member values, includ-

ing water from Eurasian rivers with different catchments and local conditions, glacial discharge, local glacier-fed rivers, and direct precipitation. The sea ice melt fraction tracer end-members also vary and in addition can have a broad range of salinities between first-year ice that has not experienced flushing with snow and surface ice melt, compared to multi-year ice which has seen one or more summers and can be completely fresh.

Water mass fractions are calculated according to a system of three linear equations (1, 2, and 3) and utilise two of the tracers

shown in Table 1.

$$f_1 \cdot x_1 + f_2 \cdot x_1 + f_3 \cdot x_1 = x_{1,obs} \tag{1}$$

$$f_1 \cdot x_2 + f_2 \cdot x_2 + f_3 \cdot x_2 = x_{2,obs} \tag{2}$$

$$f_1 + f_2 + f_3 = 1 \tag{3}$$

where f is the fraction, x is the tracer, and obs is the observation from the field. The use of different tracer combinations

has a large impact on the fractions calculated (Figure 3). This is related largely to a lack of knowledge of the end-member's seasonality and mixing history at any given time in any given point during transport rather than to measurement precision. The first column in Figure 3 shows how our data plots onto a triangle of points for each combination of end-members. The best result is obtained if the values for each end-member are unique and different at the scale of the other end-members. This presents an issue where the variables are salinity and total alkalinity since their three end-member diagrams for the end-members of

Atlantic Water, sea ice melt, and meteoric freshwater, is close to a straight line (Figure 3 a). This leaves $\delta^{18}$O-S (Figure 3 d), and $\delta^{18}$O-TA (Figure 3 g) and for meteoric water (Figure 3 h), these produce similar fractions with a difference between them of only -0.02 ± 0.01. For sea ice melt (Figure 3 i) however, the predicted fractions are very different. The $\delta^{18}$O-S predicting a far smaller brine (negative sea ice melt) fraction than that ccalculated using $\delta^{18}$O-TA. It is not possible to determine which of the latter has a lower error associated with it since we do not know the precise history of the water measses measured. This

is different than was found by Yamamoto-Kawai et al. (2005) who found sea ice melt to be predicted with a difference of 0.03





**Table 1.** End member values used to determine water mass fractions. Meteoric water values for $\delta^{18}$O and TA are those of the Lena river. Sea ice melt values for $\delta^{18}$O and TA are from own measurements on the shelf

|  | Salinity | $\delta^{18}$O (‰ VSMOW) | TA (µmol/kg) |
| --- | --- | --- | --- |
| Sea ice melt [a] | 2 | -2.344 ± 0.746 | 204 ± 162 |
| Meteoric [b] | 0 | -20.5 | 788 |
| Atlantic [a] | 34.9 | 0.32 ± 0.34 | 2267.33 ± 50 |

a. Own data

b. Cooper et al. (2008)

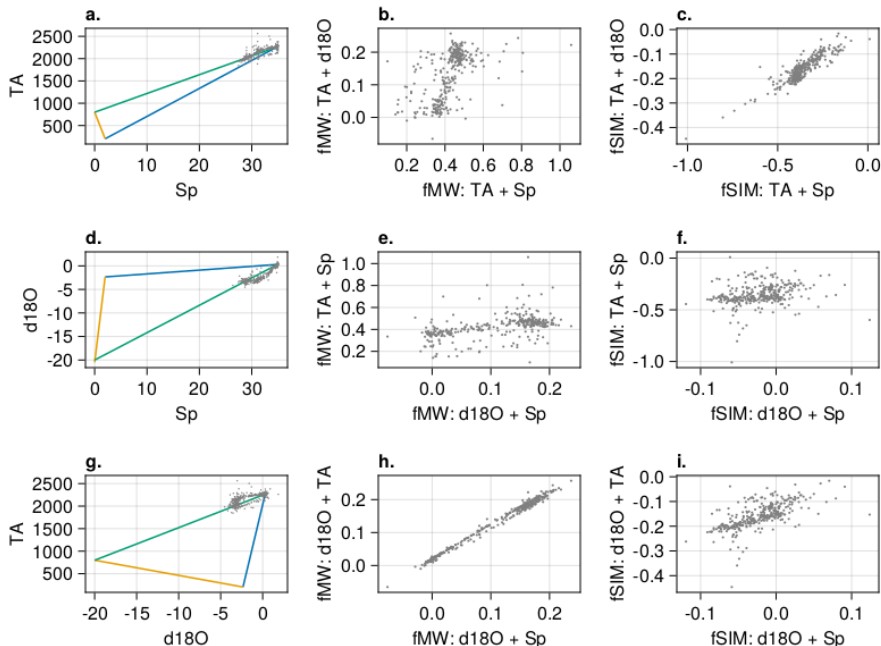

**Figure 3.** A comparison of different 2 end-member fraction calculations showing how much the choice of tracer can influence the fraction calculated for each end-member type. The mixing triangles are shown in the first column, meteoric fraction in the second column, and sea ice melt fraction in the third column. The coloured lines connect the different end-members of Meteoric (MW), Sea ice melt (SIM), and Atlantic Water (AW) masses. Green is AW - MW, blue is AW - SIM, yellow is MW - SIM

for most their data using salinity with TA versus salinity with $\delta^{18}$O. Compared to our data where the difference in sea ice melt fraction using these tracers are a magnitude larger at 0.33 ± 0.1 (Figure 3 c), and using $\delta^{18}$O with either TA or Sp (Figure 3 i) 0.14 ± 0.04.



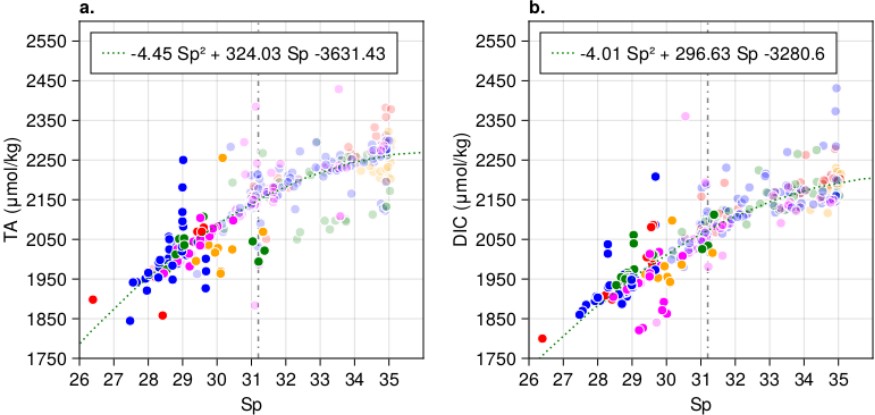

**Figure 4.** Variation of the measured carbon system parameters with salinity and their polynomial fit. The vertical line at S=31.2 highlights the remnant of the winter mixed layer water as described in (Willcox et al., 2023). Opaque values are mixed layer data based on $N^2$, transparent values are below the mixed layer

## 2.4 Salinity normalization of carbonate system measurements

To analyse changes in TA and DIC that are not conservative with salinity such as changes due to (de-) nitrification, sulfate reduction, ikaite precipitation and/or dissolution or primary productivity, the influence of salinity needs to be removed from the data. Several approaches are commonly used in the literature. These are frequently variations based on the original formulation,

$$nX = \frac{X_{meas}}{S_{meas}} \cdot S_{ref} \qquad (4)$$

where X is the variable to be corrected for, e.g. TA and/or DIC, Sp is the salinity, and meas and ref subscripts stand for
the field measurements and the reference value respectively. Later iterations include corrections for a TA estimated by linear regression at the point Sp = 0 (Friis et al., 2003), or correct for the calculated sea ice melt fraction (Yamamoto-Kawai et al., 2005). Each of these corrections has associated issues and errors and may not provide useful information, especially where there are multiple low salinity sources for TA such as shelf environments host to catchments with differing geology. Furthermore, the use of a reference salinity is arbitrary. Although there are more official description of what a reference salinity is (Wright
et al., 2010), in the case of salinity normalisation they are generally chosen to be the highest salinity seen for the dominant watermass in a particular region. This makes any comparison between different geographical regions with a difference in most saline input and therefore chosen reference salinity for calculated values subject to bias. This complexity primarily impacts mixed layer depths (Friis et al., 2003) where the meteoric-influenced layer is highest. If these normalizations rely on other assumptions such as those underlying the calculation of sea ice melt fraction from $\delta^{18}$O, any error in these assumptions will
be propagated into any subsequent application using the normalized data.





The processes controlling the water mass composition and the associated shelf salinity and alkalinity are complex. In addition, fraction calculations suffer from the ambiguities discussed in the previous subsection, therefore these data might best be normalized with respect to salinity by the simple removal of a polynomial-predicted value from the data, rather than attempting to correct for the assumed representative values for the Northeast Greenland shelf which contains such vastly variable sources in unknown relative quantities. For purposes of comparison and to choose the best representative method for the salinity normalisation of the carbonate system data, four different salinity corrections were applied (Figure 5). The first (Figure 5a) is the direct application of the polynomial from Figure 4,

$$X_{pred} = X_{obs} - X_{poly} + X_{Sp,max} \tag{5}$$

where pred is the salinity-normalised value estimated by the equation, obs is the observational data, poly is the value predicted by the polynomial according to the equations in Figure 4 a, and $X_{Sp,max}$ is, in this case where the variable to be normalized is TA, the average TA measured at salinites larger than 34.9. This method therefore still relies on an arbitrary choice of reference salinity but it reduces the number of assumptions made about the data, and rather makes the assumption that the polynomial captures the water fractions.

We compare the polynomial corrected values to the traditional correction according to Equation 4 (Figure 5 b), the sea ice meltwater fraction only correction (Figure 5 c), and the normalisation according to Friis et al. (2003) (Figure 5 d), none of which entirely remove the influence of salinity on the data. The only two normalisations which produce a result relatively free from any pattern with respect to salinity are the direct application of the polynomial (Figure 5 a) and the sea ice correction followed by the correction pioneered by Friis et al. (2003) (Figure 5 e), which in this case could be considered a correction for the input of meteoric water. The linear relationship with a slope close to 1 in (Figure 5f) between these two ways of normalising the data with respect to salinity show that these might be used interchangeably, particularly where no $\delta^{18}O$ data is available in a system dominated by the admixture of sea ice melt and meteoric water to Atlantic Water. Any correction method introduces bias in the resulting normalized data. We choose to use the simplest correction with the least assumptions in an effort to minimize this bias i.e. the correction using a fitted polynomial (Figure 5a).

## 3 Results

Comparison of the total alkalinity (TA) and dissolved inorganic carbon (DIC) in the top 150 m between the TS-groups shows some intriguing differences (Figure 6). Surface values of carbonate system parameters have a large variability, ranging between ~1800 μmol/kg (Slope 2 and Main Shelf groups) and ~2300 μmol/kg (Main shelf and Off Shelf groups) in TA and 1800 μmol/kg to 2100 mol/kg for DIC. The median full depth $pCO_2$ for each TS-group is Off Shelf: 611.8, Slope 1: 513.17, Slope 2: 481.1, Main Shelf: 412.31, and North Shelf: 367.44 and for the mixed layer only: Off Shelf: 611.6, Slope 1: 504.7, Slope 2: 485.9, Main Shelf: 391.9, North Shelf: 358.4. Surface layer values are lower on average for each TS-group (column) with the most distinct difference between above and below the mixed layer (as determined by depth of maximum $N^2$) in the Off Shelf and

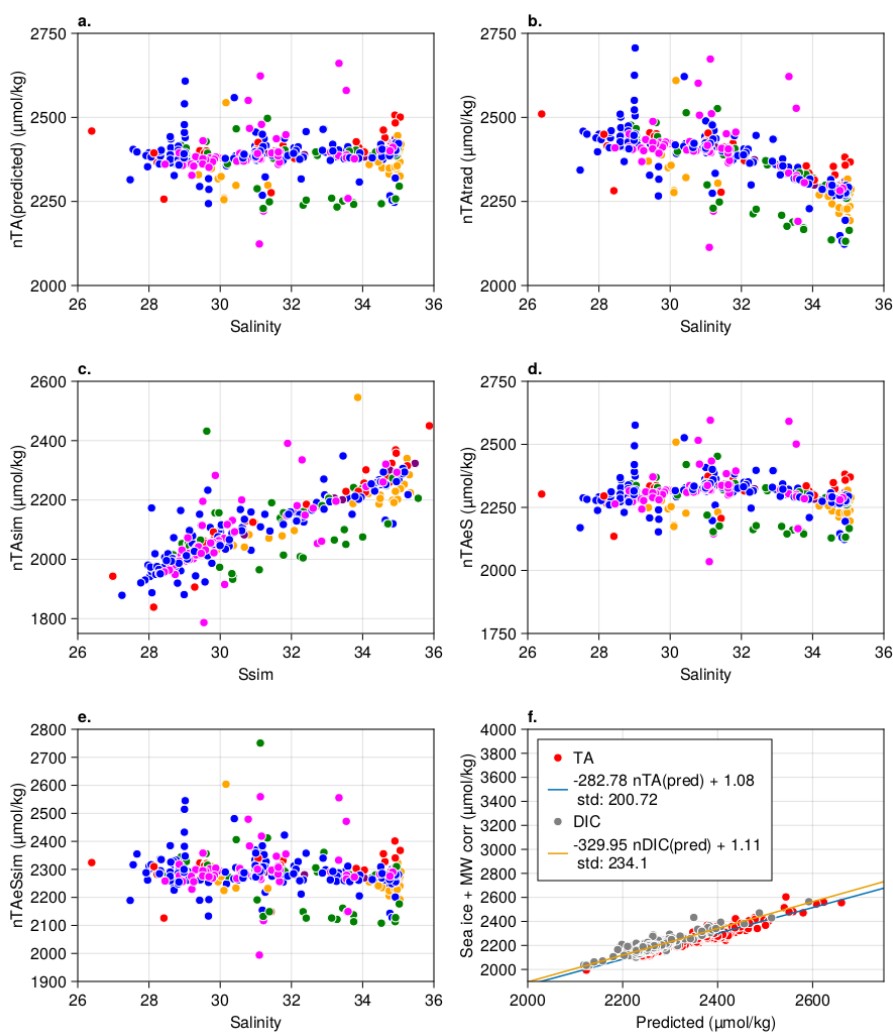

**Figure 5.** Comparison between different ways of salinity normalising the data. Normalisation using the polynomial from Figure 4a, traditional normalisation (b), sea ice fraction correction (c), meteoric water correction with TA calculated for S=0 per Friis et al. (2003) (d), sea ice correction first, then subsequent meteoric water correction (e), and a comparison between the polynomial correction and the stacked sea ice and meteoric corrections for both measured carbon system parameters (f).





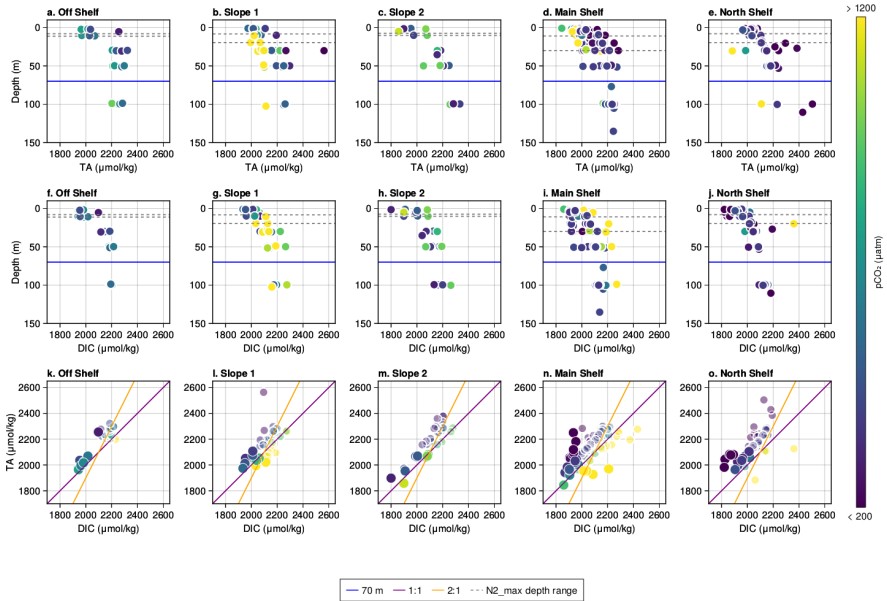

**Figure 6.** Observations for each group. First row shows TA with depth (a to e), second row DIC (e to i), and the last row TA against DIC. Colors represent $pCO_2$, opaque values are the mixed layer and the transparent values below the mixed layer.

North Shelf TS-groups. The broadest range of surface values is found on the main shelf where most measurements were obtained. Though much of the calculated $pCO_2$ on the shelf is higher than the atmospheric partial pressure of the gas (> ~400 ppm.), extreme elevations are achieved where the TA to DIC ratio drops below 1 (bottom row in (Figure 6). This seems most prevalent in the Slope 1 and Main Shelf TS-groups. The North Shelf (located in the Northeast Water Polynya region) has the lowest calculated average $pCO_2$.

## 3.1 Influence of sea ice melt on $CO_2$ partial pressure

Both TA and $\delta^{18}O$ are independently influenced by sea ice formation. Much brine is lost during initial sea ice formation, especially in 'latent heat' polynyas where frazil ice formation is dominant such as the flaw polynyas on the Siberian shelves in the source regions of the Transpolar Drift. Much TA will be lost simultaneously unless it is captured into the ice matrix where it may form the mineral ikaite ($CaCO_3 \cdot 6H_2O$) (Rysgaard et al., 2013). Meawhile the $\delta^{18}O$ fractionates due to the heavier water being preferentially taken up into the ice matrix (Souchez and Jouzel, 1984).

The $\delta^{18}O$:TA is characterized by a triangular pattern between points 1, 2, and 3 (Figure 7 a,c) which is not found in the $\delta^{18}O$:DIC (Figure 7b,d). TA is considered to be conservatively mixed where DIC is not and the data between point 1 and two are interpreted as falling on the Atlantic Water to meteoric water mixing line, where the data betweeen 2 and 3 are considered to represent meteoric to sea ice influenced water. The most extreme values of $pCO_2$ are primarily associated the Slope 1 group



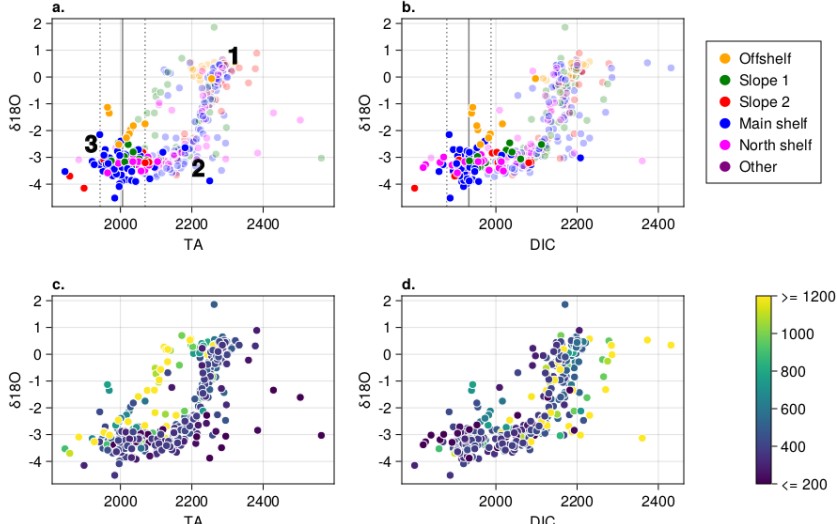

**Figure 7.** Carbon system parameters plotted against stable water oxygen isotopic composition and coloured by group (a,b) and calculated $pCO_2$ for all data (b,d). mixed layer data are opaque. Higher and Lower $pCO_2$ indicate relative values, concentrations in c are a wider range than in d. The grey lines in a and b are average TA (2006 ±63) and DIC (1933 ±54) for the Main Shelf group mixed layer only

and fall between point 1 and 3 with lower than average (by almost 200) TA compared to meteoric-influenced water and higher $\delta^{18}O$. These data have either average or higher than average DIC.

## 3.2  Salinity normalized data

Salinity-normalized data can be used to determine dominant processes controlling the carbonate system (Zeebe, 2012) and whether these vary by group, geographical location, and/or nearby features. The Pearson's correlations between the $pCO_2$ for each group in the mixed layer and the salinity-normalized carbonate system parameters as well as several directly measured parameters are shown in (Figure 8). Immediately obvious is that $\delta^{18}O$ and meteoric water fraction are the exact (±0.01) inverse of one another for each group as also shown in (Figure 3). What is surprising is that temperature has no to weak correlation to calculated $pCO_2$, particularly on the main shelf, and excludes temperature as a dominant control on the gas.

It is apparent that each group is distinct from one another particularly in terms of the measured and normalized carbonate system parameters, with an anti-correlation (negative) relationship between TA and $pCO_2$ that is strongest in the Off shelf group. The DIC shows the opposite trend with a very high correlation between DIC and $pCO_2$ in the Main and North Shelf areas where no other parameters seem to have any influence.

In the Off Shelf water the TA and DIC are both negatively correlated to the $pCO_2$. This does not seem to be related to the Atlantic Water fraction present or any other known variable except for potentially it's geographic location, with the Off Shelf group $pCO_2$ showing a moderately significant anti-correlations to latitude and longitude. The Slope 1 group stands out in terms





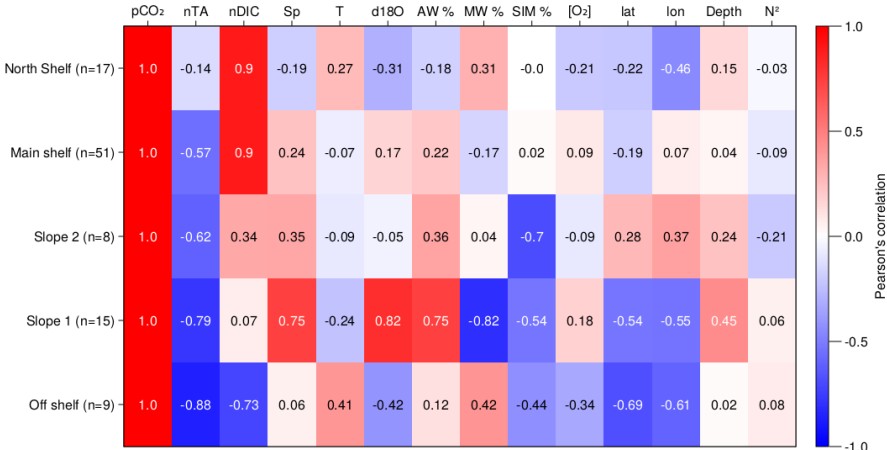

**Figure 8.** Pearson's correlation table for $pCO_2$ for any controls thought to be possible influences on $pCO_2$ on the shelf. Each row represents a TS-group, each column the correlation to $pCO_2$. Strongest positive correlation is +1, strongest negative correlation is -1 and no correlation is 0. We consider moderate significance for values $|0.5 < r < 0.75|$ and strong significance for values $|r > 0.75|$

of the strongest correlations between the water mass fractions through salinity and $\delta^{18}O$ and $pCO_2$. Slope 2 data are primarily anti-correlated to negative sea ice melt fraction (insinuating a positive correlation with brine).

Although relationships with latitude and longitude vary and are stronger along-slope than on the shelf itself, plotting the normalized data against latitude provides us with an approximate location for possible geographically constrained influences such as large glaciers and fjords or bathymetric features (Figure 9). It can be seen that above 79N within the Northeast Water Polynya, the TA in the mixed layer is average for the dataset however the DIC has several lower than average outliers. Conversely, Young Sound is associated with a peak in TA, as does the region near the exit of Dove Bugt which receives meltwater and ice from the surge-type glacier StorStrømmen, though this area also has some low values (Figure 9 a). The DIC is high on the slope (Slope 1 and 2 TS-group) between 78 and 79 °N (Figure 9 b). Plotted against one another, salinity normalized values can be indicative of the dominant processes on the shelf (Figure 9c). The highest mixed layer TA associated with Young Sund are steep even for $CaCO_3$ dissolution and might be influenced by geology which can't be shown by this type of plot (Figure 9a). Both $CO_2$ uptake and release seem to be taking place with the latter associated with the North Shelf and possibly with primary productivity. Mixed layer waters in the Off Shelf group and the higher $pCO_2$ waters of the Slope 1 group just below the mixed layer are associated with the formation of $CaCO_3$ though the dominant process causing the formation of the mineral remains elusive from this plot.

### 3.3 Atmosphere-ocean exchange of $CO_2$

The mixed layer is the depth available for possible source or sink behaviour with rescpect to $CO_2$ between atmosphere and ocean and, as determined by maximum $N^2$, is shown in (Figure 10) together with the mean $pCO_2$ found between the surface



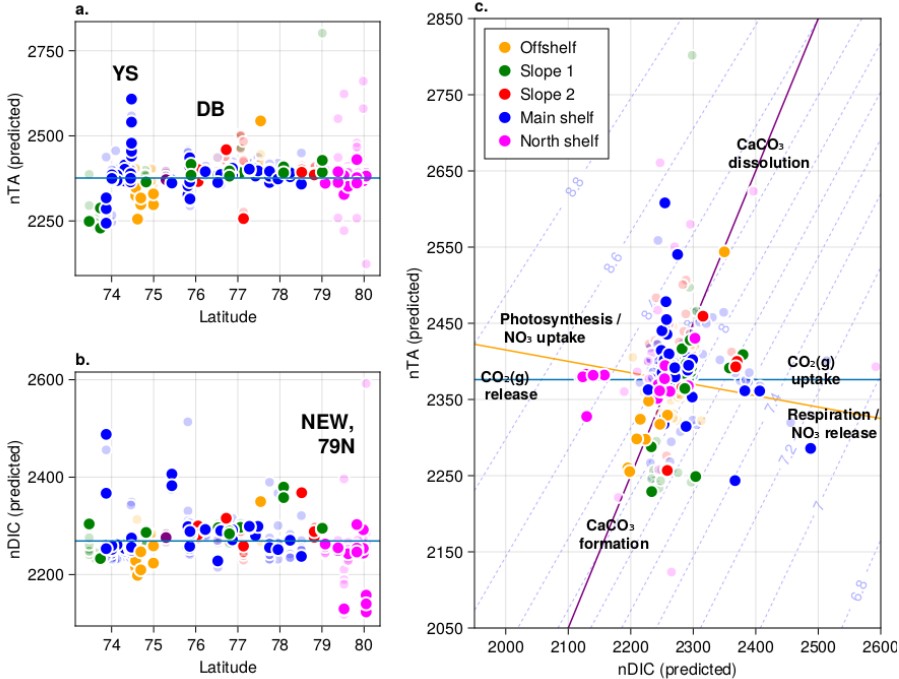

**Figure 9.** Double normalised (sea ice fraction then salinity) TA (a) and DIC (b) data variation with latitude and one another (c) is TA against DIC. Contours generated by CO2SYS at a temperature of -2 °C and a salinity of 31 to approximate representative values for the surface mixed layer on the shelf. Transparent and opaque points are all data versus mixed layer only respectively. YS refers to Young Sund, DB for Dove Bugt which receives meltwater and ice from Storstrømmen glacier, NEW to the Northeast Water Polynya, and 79N to Nioghalvfjerdsbrae

and this depth for each station. The stratification is highly variable and only partially explained by distance to the coast and/or slope. Average mixed layer $pCO_2$ in bathymetrically connected regions such as Belgica Trough apparently decreases with decreasing distance to the coast which may reflect the influence of the known counter-clockwise surface current in this location (Budéus et al., 1997). The region around the mouth of Young Sund similarly shows a possible cross-shelf trend in $pCO_2$. In the mid-shelf region between the southern Young Sund transect and Belgica Trough there are fewer patterns with values above and below atmospheric partial pressures present and in no discernible order although mixed layer depths are shallower toward the coast and thus can reasonably be expected to limit atmospheric exchange.

# 4 Discussion

Some assumptions are required to conceptually simplify the processes dominating the carbon system on the Northeast Greenland shelf and enable their analysis. Since these potentially introduce systematic errors, these will be discussed prior to discussing the data.





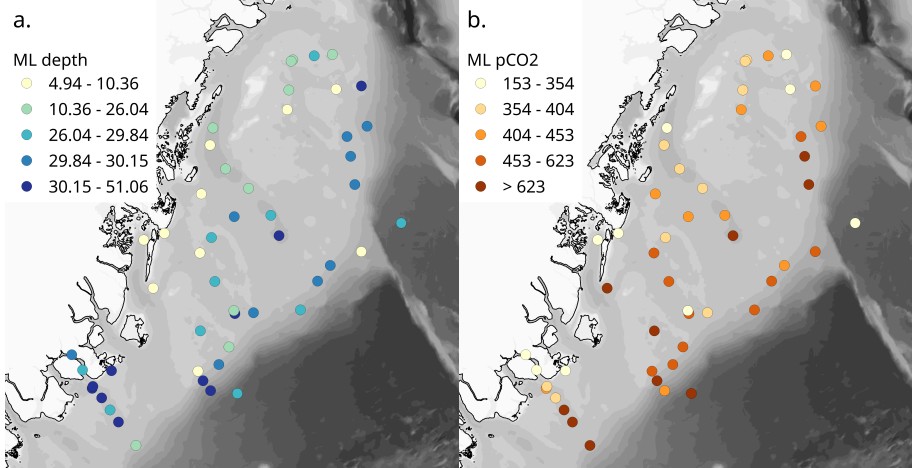

**Figure 10.** Maps showing the depth of the mixed layer according to depth < depth $N^2$ max $pCO_2$ (a) and the $pCO_2$ averaged across these depths (b) to determine the potential average release or uptake of $CO_2$ to the atmosphere during the time of sampling

## 4.1 Northeast Greenland shelf alkalinity

Alkalinity is considered conservative with respect to salinity in the open ocean. Therefore in an ideal 2 end-member environment, alkalinity is expected to vary linearly with salinity between a single ocean source and a single freshwater source providing the non-conservative constituents are negligible. The Northeast Greenland shelf receives water from many different freshwater sources, each of which can leave its own imprint on measured alkalinity values and would increase systemic error if not addressed. Some of the alkalinity sources and processes are shown in (Figure 1a).

The alkalinity of a terrestrial freshwater source depends on its substrate and rate of weathering of $CaCO_3$ (Millero, 2013). This, together with any subsequent biotic or abiotic changes, means each river that discharges from a different catchment can have a vastly different freshwater alkalinity end-member, as is certainly the case for the six largest Arctic rivers (Cooper et al., 2008). Because the Northeast Greenland shelf is one of only two major outflow regions of the Arctic Ocean, the other being the Canadian Arctic Archipelago, multiple sources can be expected to be present on the shelf.

Rivers also supply large volumes of terrestrial dissolved organic matter which can impact alkalinity by acting as a proton acceptor (Middelburg et al., 2020). In addition it can be photo-oxidized, releasing $CO_2$ (Bélanger et al., 2006) though this effect may be small due to the low solar angle at high latitudes. Contributions to alkalinity by (fluorescent and/or dissolved) organic matter can no longer be discounted (Kim and Lee, 2009; Kerr et al., 2021), particularly where water masses are subject to production of fluorescent organic matter that is associated with sulfate reduction and enhanced alkalinity such as within the Chukchi shelf sediments (Chen et al., 2016). These sediments are already known to contribute to such signals as the denitrification-influenced nitrate to phosphate ratio, which is frequently used as a tracer for waters passing across these sediments (Jones and Anderson, 1986,  ; Jones et al., 2003) and are also known to take place in other regions such as the East Siberian and Laptev Seas (Nitishinsky et al., 2007). River-associated coloured dissolved organic matter is present on the



Northeast Greenland shelf (Stedmon et al., 2015) and is thought to play a role in heat absorption (Granskog et al., 2015). Both heat absorption and any reducing processes within this layer or in its formation region may influence the carbonate system by decreasing gas solubility and changing the alkalinity.

Besides the aforementioned sea ice melt another potential crysospheric source of alkalinity is glacial discharge, with the largest contribution of the latter likely associated with marine-terminating glaciers that scour across beds with a high carbonate content. We do not at present have any direct measurements of individual Northeast Greenland glacier discharge alkalinity though much of the bedrock in this region is dolomite and is likely to contribute calcium and magnesium to runoff (Smith and Moseley, 2022). The contribution to the total (titration) alkalinity from glaciers could also be low if it is sourced primarily

from the supraglacial environment, e.g. snow, firn and/or ice melt that has not been in contact with bedrock since precipitation has an alkalinity of 0 and there are no known englacial alkalinity sources. A lot of the subglacial freshwater signal is thought to be lost fairly quickly during transport away from the source (Beaird et al., 2015; Mortensen et al., 2020) and therefore any enhancement in alkalinity may be highly localized. We do have total alkalinity values measured in the glacier-fed rivers that are discharging into Young Sund (Sejr et al., 2011) though whether the observed amounts of annual alkalinity contribution

(e.g. 440 μmol/year) is enough to explain the spike observed outside the fjord on the shelf and along the slope is uncertain 9.

We know that water on the Northeast Greenland shelf is subject to denitrification since this signal is frequently used as a tracer (Jones et al., 1998; Falck, 2001; Jones et al., 2003; Falck et al., 2005; Dodd et al., 2012). Denitrification can influence alkalinity (Middelburg et al., 2020) though this influence may be so small that it is negligible.

The geochemical signal of the meltwater from the Greenland Ice Sheet is small (1.8% at the glacial front according to

260 Huhn et al. (2021)) compared to the volume of freshwater transported by the Transpolar Drift which is readily identifiable for example using coloured dissolved organic matter (Granskog et al., 2012). If the different upstream sources of alkalinity (Figure 1a) are well mixed prior to their advection onto the Northeast Greenland shelf, they can be treated as a single source by determining the alkalinity of the combined sources from linear regression against salinity and solving the equation for a salinity of zero according to the second equation in Friis et al. (2003). The shelf is highly stratified with a thin surface mixed

layer that is heavily influenced by sea ice melt and freeze processes, and the TA:S plots are more appropriately fitted with a polynomial rather than a straight line from which a salinity of 0 may not give a combined meteoric source alkalinity for the water being advected from the Arctic (and/or North Atlantic) ocean.

### 4.2 Regarding water mass fraction calculations

The use of linear equations to determine source fractions in Arctic ocean outflow regions is an increasing subject of debate.

For example, Forryan et al. (2019) determined that sea ice brine and melt fractions sum to net zero across all the Arctic ocean exit gateways. When sea ice brine is formed, it sinks into the winter mixed layer or is transported even deeper through dense water cascading until it reaches a depth of neutral buoyancy (Ivanov and Golovin, 2007; Luneva et al., 2020). These brine-enhanced layers can be transported in a different direction and exit via a different gateway than the sea ice it was drained from and take different amounts of time even where they do leading to potential inter-annual and seasonal differences in fraction to

the original water mass the ice formed from and resulting in a non-representative water mass fraction for sea ice melt and/or




brine (fractions don't sum to 1). As a result, sea ice melt and sea ice brine cannot simply be assumed to be net zero within each individual region or gateway. The determination of melt and brine fraction therefore require a good understanding of the properties of the water that these processes influence directly in their source region and season, rather than making inferences of a local balance at individual exit gateways at a given time

### 4.3 TA and DIC on the Northeast Greenland shelf

Differences between our data and that collected in the previous decades are immediately apparent. Whereas previous studies classified the region as a $CO_2$ sink (Yager et al., 1995; Takahashi et al., 2002; Jeansson et al., 2008), our data show that at least during the period of our observations the area can act as a $CO_2$ source, particularly in certain TS-group (Figure 6). Previous studies focused mainly on the Northeast Water polynya dominated area, north of Belgica Trough and may not be representative for the rest of the shelf. South of Belgica Trough, the highest calculated $pCO_2$ fall below a TA to DIC ratio of 1:1 (Figure 6k-o). This effect is particularly apparent for the Slope 1 group, although this pattern can also be seen in Main Slope stations. Although this is not the entire story with respect to these high calculated $pCO_2$ values, since the Off Shelf group shows the highest median $pCO_2$ across all depths, the mechanism whereby either TA is lost or DIC is gained across the shelf requires investigation.

### 4.4 Alkalinity, meteoric water, and sea ice melt

One proposed mechanism for the original conclusion for the region being a $CO_2$ sink is the presence of large volumes of sea ice melt in the (surface) mixed layer. Generally the determination of sea ice melt fraction utilises the fractionation of $\delta^{18}O$ (Östlund and Hut, 1984) or incorporation of TA (Tan et al., 1983) combined with a loss of salinity to the underlying water column during seawater freezing as a 3 equations with 2 known variable linear system. Further these two data can be plotted against one another (Lansard et al., 2012; Mol et al., 2018) to differentiate between meteoric and sea ice melt influenced water in freshwater influenced Arctic shelves. Our data do not show the same clear relationships with lower $pCO_2$ in the Polar mixed layer and higher values deeper in the cold halocline layer, particularly enhanced in the Upper Halocline. In addition, their data do not show a similar triangular pattern in TA, missing an equivalent to our values between points 1 and 3 (Figure 7a).

The anomalously low salinity-normalised TA in all TS-group except for the North Shelf group and high or average salinity-normalised DIC in the Main and North shelf TS-group are primarily responsible for the highest $pCO_2$ calculated (Figure 7 and 8). Of course since these non-normalized data were used as inputs the calculated $pCO_2$ values are not independent.

We identified four possible drivers for these anomalies in TA:

1. Mixing with a secondary water source, either a freshwater source with a different TA for salinity = 0 or a secondary saline water source with a different TA for Sp > 34.9

2. Injection of brine into the system causing lowering of TA and increasing DIC

3. Biogenic calcification which decreases the TA while maintaining the same salinity.



    4. Mixing with a water source that passes near/through an alkalinity titrating hydrothermal vent system which removes TA through the formation of $CaSO_4$

Mixing with a low TA water source would require the identification of a plausible water source. Glacial water with a lower

(by 30 μmol/kg) TA than DIC has been observed in West Greenland (Meire et al., 2015). The latitude of Young Sund, just north of the region of low TA, instead shows a spike in TA and Dove Bugt doesn't have values low enough to explain the observations. The Off Shelf water between latitudes of 74 and 75 °N could be a plausible source. Off Shelf water is characterized by the absence of a cold halocline layer (Willcox et al., 2023) which means it it composed of Atlantic Water and surface water fractions. Neither of these are commonly associated with low TA therefore a biogeochemical change is still required before or

after mixing (Jiang et al., 2014).

The formation of brine as part of the seawater freezing process is expected to lower TA due to ikaite formation inside the ice, while losing DIC conservatively with salt. Most brine is rejected during frazil ice formation rather than in more consolidated ice, which loses more if it freezes more slowly. Although supercooling of several mK (Lewis and Perkin, 1983; Dmitrenko et al., 2010; Ito et al., 2020) has been observed in polynyas and therefore could hypotetically be cold enough to form ikaite

crystals in open water, this has not been directly observed. Our normalised carbon system values (Figure 9c) do indicate a strong role for the formation of $CaCO_3$. This could be ikaite from a different source than sea ice such as through reactions of seawater calcium with sedimentary carbon as has been observed on an Antarctic shelf (Suess et al., 1982) however cold temperatures or high pH (Tollefsen et al., 2020) are a pre-requisite for formation and therefore another mechanism may be more likely.

Extremely high primary productivity of calcium carbonate precipitating organisms can lead to a perceivable lowering of $CaCO_3$ without reducing salinity. Such lowering of TA with a stable salinity has been observed in both the red sea (Jiang et al., 2014) and the Bay of Biscay (Suykens et al., 2010) though never for the Arctic. The organisms that are generally associated with $CaCO_3$ formation include coccolitophores, foraminifera, and pteropods but may also include marine bacteria (Heldal et al., 2012). Coccolithophores have been shown to thrive in regions where water from the Arctic Ocean mixes with Atlantic

water (Dylmer et al., 2013) but their presence or absence cannot be verified by our data. A previous study of the primary producers in the region did not find any coccolithophores (Krawczyk et al., 2015) though several different assemblages were found associated with the different water masses. The very warm water (up to 6°C) found in 2017 (Willcox et al., 2023) was not present in their study.

Another mechanism that has the potential to lower alkalinity are the formation of $CaSO_4$ and the acidification of water

through the increase in $H^+$ associated with the percolation of water through hydrothermal vent systems (German and Seyfried, 2014). Two hydrothermal vent complexes were identified in proximity to our study area (Rysgaard, 2018) and therefore these process cannot be entirely neglected as playing a role though no sulphate or (trace) metal measurements were made during these cruises with which to draw this conclusion.



## 4.5    Evaluating present state of the seasonal rectification hypothesis

Until the influence of warm ocean water on the melting of tidewater glaciers and ice streams became apparent (Straneo et al., 2010) there were limited primary source observations available for water on the Northeast Greenland shelf. Though bottle data (TA and/or DIC) samples were collected by cruises starting with the USCGC Westwind (1979), USCGC Northwind (1984), USGC Polar Sea (1992), RV Polarstern (1993, 1999), and IB Oden (2002), sampling for these parameters was limited. The first two cruises collected only total alkalinity (TA) data to be used as a tracer for sea ice (Tan et al., 1983). Yager et al. (1995)

compiled and analysed TA and dissolved inorganic carbon (DIC) from 1992, Daly et al. (1999) used the 1999 cruise data to analyse the Redfield ratio and presents only DIC measurements, and Jeansson et al. (2008) described both the TA and DIC from 2002. More recently TA and DIC data for Northeast Greenland fjords were presented by Henson et al. (2023) who compared these with fjords along the western Greenland shelf and concluded that high dilution by freshwater sources drives acidification inside the fjords. The coverage on the Northeast Greenland shelf of these datasets is concentrated along Belgica Trough,

within the Northeast Water Polynya, and along the shelf edge and slope. The main shelf area has received comparatively little attention. Several ships have measured $CO_2$ directly, either by using gas chromatography or underway infrared measurements. These can be found in the SOCAT database (Bakker et al., 2023) but may not be easy to compare to our data due to differences in results between TA/DIC and $pCO_2$ direct measurements where previously compared (Sejr et al., 2011).

The general consensus from these previous studies is that the Northeast Greenland shelf is a $CO_2$ sink though by how

much is still under debate (Jeansson et al., 2008). This is based on two concepts, the first of which is the seasonal rectification hypothesis which states that the polynya experiences strong productivity driven drawdown in the open water season and is ice-covered once the system becomes respiration dominated and would start to release $CO_2$ to the atmosphere, and the second the idea that the shelf receives such large volumes of sea ice and its melt that the (compared to DIC) enhanced alkalinity should enhance the uptake of atmospheric $CO_2$.

Our data confirm that the North Shelf group mixed layer acts as a sink with respect to $CO_2$ though the region is not representative of the rest of the shelf which shows high variability between stations. The North Shelf $pCO_2$ is driven by low values of DIC rather than enhanced TA (Figure 9b) in the mixed layer so this likely reflects either $CO_2$ release or primary productivity. In the mixed layer across Belgica trough where there is a clockwise current toward the coast, there is a sequential decrease in $pCO_2$ with proximity to the coast (Figure 10b) that almost matches with a reduction in mixed layer depth from >

30 m to < 10 m (Figure 10a). This is also the region with the most persistent sea ice cover, which may be related to the thinning of the surface layer (Figure 2).

The Main Shelf mixed layer has an median $pCO_2$ value very close to that expected for the atmosphere and may act as either a source or sink interchangeably. Whether the Main Shelf acts as a source or sink depends on a combination of geographic location, eddies, wind fields, and other physical mixing mechanisms, and biological interactions that this study does not have

the sampling density to identify.

The mixed layer along the slope is deeper than that on the inner shelf on average, and associated with that are higher median in calculated $pCO_2$ (Figure 10). Both the Slope TS-groups show that the impact of low TA is dominant in the calculated $pCO_2$



(Figure 8) compared to that of high DIC. In the case of the Slope 1 group this seems to be driven by strong physical mixing as shown by measured salinity and $\delta^{18}$O, and the calculated water mass fractions where for the Slope 2 group the highest correlation is with the calculated sea ice melt fraction. The Off Shelf group has the highest median values for pCO$_2$ in the mixed layer and is certainly a source to the atmosphere during September 2017.

The complexity that is shown by our dataset is much higher than could be anticipated by previous studies in the region. Our 2017 measurements occurred in exceptional circumstances since the summer was extraordinarily warm and had anomalously low sea ice. Yager et al. (1995) discussed that their surface (< 70m) & polar water total alkalinity values correlated with salinity, which ours do not. The proposed seasonal rectification hypothesis is still a contender for the North Shelf region, however there was no ice inhibiting $CO_2$ exchange with the atmosphere on most of the shelf while clearly the surface values acted as a source in multiple locations. Whether this is indicative of a change which occurred in the intervening decades, or predictive of a change yet to come is currently unclear and requires further study and monitoring. It may also simply be typical for the region during this time of year. Since we have such a dearth of measurements, it is impossible to determine how much of our measurements represent change.

## 5   Summary

Our data show a clear departure from previous measurements made on the Northeast Greenland shelf and put into question whether the region acts as a consistent $CO_2$ sink. Upstream modification to the total alkalinity (TA) of the water advected onto the shelf from the Arctic Ocean as a result in water mass source changes through a shrinking or expanding of the Beaufort Gyre and it's freshwater retention may be responsible for some of the observed differences. Locally, much of the variability in the carbon system is located in the surface mixed layer on the shelf proper and modified through processes taking place both there and through slope and shelf edge interactions. Any additional detail such as from the influence of sea ice melt lenses or eddy formation require a higher spatial and temporal sampling density across more of the shelf. The timing and magnitude of respiration and the mixing of denser water masses with higher concentrations of DIC and/or lower concentrations of TA toward the surface are important in determining whether or not the region remains an annual source or sink with respect to $CO_2$. Many more direct observations of this region will be required to adequately validate this region as an annual net source or sink and provide a baseline by which to measure any future change. This determination will also require an understanding of the complex processes incluencing properties which are usually considered conservative but may change on the Northeast Greenland shelf, e.g. the TA.

*Data availability.* We are currently involved in adding the data to the Pangaea data repository and will make the set available as soon as they have accepted it



*Author contributions.* Marcos Lemes and Mikael Sejr performed the geochemical laboratory measurements for TA and DIC in their institutes respectively. Subsequent data analysis, writing of code, and initial drafting of the manuscript was performed by the primary author. Extensive feedback on first and second drafts of the manuscript was obtained from all co-authors.

*Competing interests.* The authors declare no competing interests.

*Acknowledgements.* We would like to acknowledge the contributions of the captain and crew of RV DANA for theirexcellent assistance during our field cruise to NE Greenland. We would also like to thank Egon Frandsen for support on logistics and operations.



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
