# Peer review of "The Northeast Greenland shelf as a late-summer $\mathrm{CO}_2$ source to the atmosphere"

_EGUsphere, 2024_

## Referee Comment (RC1)

**Comments on** *"The Northeast Greenland shelf as a late-summer CO2 source to the atmosphere"* **by Willcox et al.**

**Summary**

The manuscript by Willcox and colleagues presents a novel approach combining biogeochemical and physical oceanographic data to address the question whether the Northeast Greenland shelf (NEGS) acts as a source or sink of atmospheric $CO_2$. The authors contend that unlike commonly accepted, this area indeed acted -at the time of sampling- as a source of atmospheric $CO_2$ and discuss the major drivers which could explain these observations. Observations were made during the transition from summer to winter (i.e. transition from autotrophic to heterotrophic dominated system). Because of this, authors aim to show that their study provides insights on potential future responses of the NEGS to climate change.

**General comments**

The topic chosen by the authors is clearly relevant for a wide biogeochemical community and fits well within the scope of Biogeosciences. The manuscript is mostly well written and the quality of the presentation of the results is of high quality. Overall, the methodological approaches and data analysis are sound and the data is of high quality. Yet, the way in which the text has been structured along with some unclarities in the approaches used and the discussion, do not allow the reader to follow the connection between the gap of knowledge addressed by the study, the approach followed, its main results and overall implications.

In my opinion the major drawbacks of the study include:

- Lack of a clear definition of the study's novelty and major goals (I certainly see those, and therefore invite the authors to revise their text to show it). For instance, in the introduction the authors describe clearly and succinctly the seasonal dynamics of the area. Although the goal of the study is briefly mentioned, it does not come across why pursuing this goal is relevant and the approach followed is novel. As I understand it, the main aspect explored in this manuscript is that due to changes in the timing of the freezing-melting cycles, it is not clear whether the heterotrophic period (during which $CO_2$ outgassing is predominant) might tip the annual balance from overall sink to source. Should this be the case, this should be framed more predominantly. There is in fact a sentence in the abstract, which exemplifies well how this could be set up in the introduction: *"This is in contrast to the common perception for this Arctic outflow shelf region as a CO₂ sink during the ice-free season"*.
- The methods section strongly relies on measurements/ methodological approaches of published work by the same group in their 2023 paper. While I agree that a full description is not needed, the issue is that, at some extent, the present manuscript stops being a stand-alone contribution that could be fully reproducible without reading other sources (mainly Willcox et al., 2023). I therefore invite the authors to consider including a brief description of the sampling and other relevant aspects of the setting for the study (see comments in the attached, commented version of the manuscript).
- Although the crux of the manuscript is the discussion of source-sink dynamics of the area with respect to atmospheric $CO_2$, no air-sea flux densities of $CO_2$ are presented at all. Most of the discussion is based on calculated $pCO_2$ values which are evaluated in terms of being above or below a threshold of 400 ppm (which was not adequately justified). With the data at hand, it would be very easy for the authors to carry out an air-sea gas exchange calculation. My invitation for the authors is to do this not only because it makes sense considering the theme of their manuscript, but also because it helps putting their study

in the context of other published estimates (which of course ultimately would influence this contribution's impact on the current literature).

- The results section is quite descriptive and would -in my opinion- benefit from a merging with the discussion section. That being said, the discussion in its present form is mostly disconnected from the observations. This is why I think merging these two sections would help drawing those connections in such a way that the conclusions of the manuscript are better supported.

- The discussion is mostly focused in the comparison with regional studies and the results of the paper, as relevant as they indeed are, are not put in a larger regional context such that at the end the reader cannot grasp the impact of the data/analysis presented.

Furthermore, I noticed that there seems to be a good degree of overlap between the measurements of this study and historical data from sea surface $pCO_2$ archived at the SOCAT data base (https://socat.info/). Considering that the SOCAT data was gathered during different seasons throughout the year, I tend to think it would be enriching for the study to *a)* compare their calculated $pCO_2$ values with the of the database and *b*) use their analysis on major dominant variables of $pCO_2$ variability in the different sub-areas to provide an even more robust estimate of the $pCO_2$ source-sink changes over the seasonal cycle (i.e. using SOCAT values and the identified driving mechanisms/parameters as predictor variables to compute a regional estimate).

[Figure]

Source: https://socat.info/; accessed 18-03-2024

**Specific comments**

Additional comments to support the major points above, as well as some minor formatting issues are included in the form of a commented version of the document, which I am attaching to this assessment. I hope these are of use for the authors when revising their manuscript.

Kind regards,

Damian L. Arévalo-Martínez

[revised manuscript text omitted]

---

## Author Response (AR1)

**Supplement with CO₂ paper**

**AUTHORS**

**Contents**

**General response**

Both reviewers highlighted a need to restructure the document and indicated that the paper would be improved if it relied less on the previous paper. Furthermore, both reviewers indicated that the groups based on hydrography (termed as 'CTD groups') were inqdequately explained and that sticking to the dataset presented and focusing on factors that usually explain most of the variability in CO₂ would be preferred.

Taking this feedback into consideration we have entirely rewritten the manuscript from scratch in line with their recommendations. The revised version focuses on how the fugacity of carbon dioxide (fCO₂) as calculated by CO2SYS relates to:

- the measured variables salinity, temperature, dissolved oxygen, and chlorophyll
- total alkalinity (TA) and dissolved inorganic carbon (DIC) normalised to salinity
- calculated euclidean distances to the coast and slope (and East Greenland Current)
- sampling period

In addition we highlight that a large proportion of these data were obtained in areas where no full-depth sampling for carbon chemistry had occurred prior to this study due to perennial sea ice cover

and we compare our measurements for TA and DIC with algorithms used in literature to predict them to make our manuscript more relevant to a broader audience.

We hope our rewrite proves sufficient. The changes made are in blue.

**Changes with respect to comments by reviewer 1**

- Lack of a clear definition of the study's novelty and major goals (I certainly see those, and therefore invite the authors to revise their text to show it). For instance, in the introduction the authors describe clearly and succinctly the seasonal dynamics of the area. Although the goal of the study is briefly mentioned, it does not come across why pursuing this goal is relevant and the approach followed is novel. As I understand it, the main aspect explored in this manuscript is that due to changes in the timing of the freezing-melting cycles, it is not clear whether the heterotrophic period (during which CO2 outgassing is predominant) might tip the annual balance from overall sink to source. Should this be the case, this should be framed more predominantly. There is in fact a sentence in the abstract, which exemplifies well how this could be set up in the introduction: "This is in contrast to the common perception for this Arctic outflow shelf region as a CO2 sink during the ice-free season".
  - The reviewer is right that the goals should be stated more clearly and that whether this dataset takes place in the heterotrophically dominated part of the season is insufficiently shown in the document. The collection of the field materials that form the basis for this manuscript occurred opportunistically since the locations that were sampled are usually inaccessible due to sea ice cover. Of particular interest was to determine the $CO_2$ dynamics on this lesser studied part of the shelf and to determine the impact, if any, of low ice conditions as a type of real world laboratory for future conditions in a warmer ice-free climate. We initially assumed that the hydrographical conditions in the more extensively surveyed part of the Northeast Greenland shelf (e.g. the area near 79N glacier and the Northeast Water polynya) would be representative of the other areas of the shelf and we would be able to compare results. This turned out to not be true to the point that we wrote an entire paper to disentangle the hydrographic complexity prior to being able to analyse the carbon system. The data we present here are carbon system data for the full depth across much of the latitudinal and longitudinal range of the shelf where no prior data were previously collected. Even so, many of the processes influencing carbon dynamics on the shelf occur at smaller scales than the sampling density and therefore our ability to adequately explain all variability in the data is limited. What we can conclude is that the

pCO₂ as calculated by CO2SYS from TA and DIC are much higher than we expect them to be at this time of year. The Arctic Ocean is nitrogen limited in the surface layer [Codispoti, et al. 2013] and this is also the case on the Northeast Greenland shelf during the time of these cruises. This is likely the reason for low Chlorophyll-a fluorescence in the surface water where mixing can make it available for atmospheric exchange. Exceptions occur at lower latitudes where there is more Atlantic Water and also near sea ice (~-12 °E, ~79 °N, Figure a). Of course lack of primary productivity doesn't immediately provide evidence for net heterotrophy, especially if the concentration of organic material to remineralise is also low. The materials required to make this assessment are not available to be used in this manuscript. We will modify the introduction to state more clearly what our goal is and to highlight the importance of our findings.

– In the revised introduction we have changed focus on how our results contrinute to the larger body of scientific knowledge about the region. We have also included what data we have regarding dissolved oxygen and Chlorophyll-a fluorescence to investigate potential relationships to primary productivity. We have also made the fact that these data are from opportunistic sampling due to the absence of sea ice rather than of a coordinated prepared campaign which allowed sudden access to parts of the shelf which have previously not had full depth carbon chemistry sampling coverage.

• The methods section strongly relies on measurements/ methodological approaches of published work by the same group in their 2023 paper. While I agree that a full description is not needed, the issue is that, at some extent, the present manuscript stops being a stand-alone contribution that could be fully reproducible without reading other sources (mainly Willcox et al., 2023). I therefore invite the authors to consider including a brief description of the sampling and other relevant aspects of the setting for the study (see comments in the attached, commented version of the manuscript).

– We agree that there should be a (short) summary of the previous paper to create the context for these results. This will be added to the revised manuscript.

– We have added a few paragraphs sumamrizing the results described in the previous paper for context

• Although the crux of the manuscript is the discussion of source-sink dynamics of the area with respect to atmospheric CO2, no air-sea flux densities of CO2 are presented at all. Most of the discussion is based on calculated pCO2 values which are evaluated in terms of being above or below a threshold of 400 ppm (which was not adequately justified). With the data at hand, it would be very easy for the authors to carry out an air-sea gas exchange calculation.

My invitation for the authors is to do this not only because it makes sense considering the theme of their manuscript, but also because it helps putting their study in the context of other published estimates (which of course ultimately would influence this contribution's impact on the current literature).

- Some flux density calculations for (more coastal stations) Northeast Greenland have been previously attempted [Sejr et al. 2011, Henson et al. 2023]. A major issue is that the potential for errors here is large. No atmospheric components were measured (wind speed or atmospheric $pCO_2$) during the two cruises. Any flux calculation heavily relies on these inputs and small lacks of accuracy in these data could lead to relatively large errors. Furthermore the choice of transfer coefficient is very important and although those of Nightingale et al. (2000) we do not know whether these are representative of transfer velocities in ice-covered, partially ice-covered, and/or glacial melange dominated environments. In our opinion it is not suitable to include flux calculations in the manuscript for these reasons. We prefer to stick as close to the data itself as possible so as to not conceal any details behind potentially inaccurate calculations. However we will mention the SeaFlux atmospheric concentration for the region during the month of the cruises and update any figures where atmospheric values are noted. We have performed the $CO_2$ flux calculations as shown in Figure b using the gas transfer velocities, $CO_2$ solubility, and $fCO_2$ for the atmosphere from the SeaFlux Data Product v.2023.02 [https://zenodo.org/records/8280457].

  - We have not included flux calculations for the aforementioned reasons. We have updated and justified an atmospheric threshold of 395 µatm based on SeaFlux.

- The results section is quite descriptive and would -in my opinion- benefit from a merging with the discussion section. That being said, the discussion in its present form is mostly disconnected from the observations. This is why I think merging these two sections would help drawing those connections in such a way that the conclusions of the manuscript are better supported.

  - Agreed, the two sections will be merged in the revised manuscript

  - We have merged the results and discussion sections

- The discussion is mostly focused in the comparison with regional studies and the results of the paper, as relevant as they indeed are, are not put in a larger regional context such that at the end the reader cannot grasp the impact of the data/analysis presented.

  - We appreciate this comment. Since the region is considered not only a sink for atmospheric $CO_2$ but this water is also advected to regions where intermediate water is formed such as the Labrador Sea. Any changes in $CO_2$ uptake by the ocean in this region could influence

the longer term storage of atmospheric $CO_2$. In the revised version of the manuscript we will make sure to state the broader context of our findings.

– We have added additional sections to the introduction, results and discussion, and the summary that put our findings into broader context

- Furthermore, I noticed that there seems to be a good degree of overlap between the measurements of this study and historical data from sea surface pCO2 archived at the SOCAT data base (https://socat.info/). Considering that the SOCAT data was gathered during different seasons throughout the year, I tend to think it would be enriching for the study to a) compare their calculated pCO2 values with the of the database and b) use their analysis on major dominant variables of pCO2 variability in the different sub- areas to provide an even more robust estimate of the pCO2 source-sink changes over the seasonal cycle (i.e. using SOCAT values and the identified driving mechanisms/parameters as predictor variables to compute a regional estimate)

– During our conversations prior to submitting this manuscript, we discussed whether SOCAT data should be used for comparison. We chose not to include these for two main reasons, one is that the data are not directly comparable. The second is that SOCAT data for the region is only available for 2009 (Figure c). We will make these limitations clearer in the revised version of our manuscript, mention the SOCAT seasonal and use it to introduce the inter-annual variability for the region.

– We have added the locations of both the SOCAT data and the CARINA data in the region to the map (Figure 1a) in the manuscript, primarily to show that our data cover a region previously limited to only surface measurements (SOCAT) with our data the first full depth carbon chemistry measurements. We do not do a direct comparison inside the manuscript because it is not a substantial addition but these previous data should be mentioned so we have added a section to the supplement where we show our data ($fCO_2$ x date) and that for each year collected in those databases. The most recent September data in SOCAT is from 2009 and for CARINA from 2003.

- This abstract is very short. While this is not necessarily bad, I think in this journal the authors do have the space to include more (specific) information that allows the reader to, for instance, see the magnitudes of the source/sink terms and exchange fluxes, as well as which exactly are the proposed/observed causes for low TA values that might tip the equilibrium from CO2 uptake to outgassing.

– We will extend the abstract with more results

– We have rewritten the abstract to match the new manuscript. It is now longer

- In the paper by Willcox et al (2023) the authors state that this field work was carried out over a 3-week period in August-September 2017. The two days stated here appear too little for the number of stations included in this study. Please check and confirm the timing or change accordingly.
    - The cruise durations were a month (24 August to 25 September as stated) in total for both this and the previous manuscript.
    - No change

**Specific comments**

Due to the rewrite having replaces most of the text, comments regarding spelling for example, are no longer directly relevant

- Precipitate
    - Will change spelling
- 45: Please revise this sentence. I understand what it is meant to express, but it takes a while because it is not clearly formulated.
    - Sentence will be rephrased
- 49: "(...) describing observations of alkalinity and dissolved inorganic carbon made during (...)"
    - Will amend the grammar
- Please revise sentence as it reads odd. A suggestions would be: "(...) they provide possible insights into the variability of air-sea exchange of CO2 in the NGS under a changing climate (...)"
    - Will amend
- 56: I understand that it is not need to repeat all the methodological description for these measurements since these are published. However, the TS groups are essential information for this manuscript, which as I see it, should be understood independently of past work (i.e. without the reader to have to open a new paper to see the classification). Therefore, the authors might want to at least include Table 2 from the Willcox et al paper in 2023 as a supplementary information to this manuscript.
    - Will include a summary of major conclusions of the hydrography from the previous paper as they pertain to this document.
    - As described above, a section has been added to discuss the cruise more extensively and the results of the previous paper. Simultaneously, all mention of CTD groups has been

removed and the focus of the paper shifted toward the comparison of the variables that
should produce most of the variability in fCO₂

- 56: Introduce the abbreviation here upon first usage.
  – Will introduce abbreviation here
  – Introduced abbreviation for CTD
- 56: This is not necessary. The citation suffices in this case.
  – Will remove
  – Removed, restructured
- 59: Here it would not harm to write that samples were collected using a CTD-Rosette system.
  – Will add this text
  – Have added the words CTD Rosette in the methods
- 71: Consider a more suitable wird; e.g. "exchanges" or "fluxes".
  – The other reviewer also took issue with the wording of this sentence. This section will
    be revised to more specifically describe the difference between the actual flux at the air-
    ocean interface versus the mass available in the mixed layer which can be brought to the
    interface before the layer is at equilibrium.
  – lost in revision, referenced different paper which also used maximum N²
- 72: "(…) carbonate system data (…)"
  – Will amend
  – lost in revision
- 74: Mixed layer depths (MLD) in the region are highly variable (as indicated in the ref-
  erence cited by the authors themselves). Choosing "less than 70 m" (which according to
  Peralta-Ferriz and Woodgate 2015 is the lower limit of the winter MLD value) can there-
  fore be misleading. Unless the authors have a well-justified reason to choose this value,
  I would invite them to estimate the MLD themselves. I am convinced that this should be
  possible with the data they have at Moreover, I would like to bring the author's attention
  to at least two studies in which it has been shown that not only MLD but also near-surface
  stratification (caused by e.g. ice melt and/or freshwater inputs) has a noticeable effect
  ihttps://online.ucpress.edu/elementa/article/8/1/084/113075/Underestimation-of-surface-
  pCO2-and-air-sea-CO2f sea, Hudson Bay"
  – We are estimating the mixed layer depth ourselves using the depth of the maximum Brunt-
    Väisälä frequency squared (N²) as described. We are using 70 m as the lowest possible
    depth (cut off) at which the MLD could be found. We have evidence from the dissolved O₂
    maximum just below the depth of the maximum N² that no ventilation takes place below it.

We will change the wording of the manuscript to make this clearer. We certainly appreciate that near-surface stratification is important in the flux between ocean and atmosphere but this was not the focus of our paper.

– Sentence lost in revision. The method of mixed layer determination with $N^2$ is introduced and justified elsewhere in the document

- 76: The assumptions in this section would be reasonable if no data were available for calculating the MLD. However with data at hand, it seems to me the most sensible choice to compute the MLD instead of using a constant value.

  – This section describes how we compute the MLD, we do not use a constant value. Apparently our wording is insufficiently clear and the text will be amended to attempt to accommodate increased comprehension.

  – as above

- 85: Add a point here.

  – Will add a point

- 86: "(…) water sources(…)"

  – Will add "water"

- 119: Check for consistency; in figure 4 the axis also have "Sp", whereas equation (4) has "S".

  – Will modify the document to refer to practical salinity with only one term/acronym throughout.

  – Current document uses Salinity or abbreviated 'S' throughout

- 127: "(…) is subject to (…)"

  – This was intended as "makes [..] subject to bias" to injecting it would not work, but will rephrase and remove 'make'.

  – The entire section has been moved to a supplement

- 163: Is this from the NOAA's cooperative air sampling network? Or was it measured on board? Please clarify and if appropriate, add the corresponding data source.

  – No. At the time of writing we picked an arbitrary representative value since no local measurements are available, however we have since obtained values from the SeaFlux dataset so we will amend this where necessary throughout the manuscript.

  – We are using SeaFlux values for atmospheric concentrations now

- 164: It would be useful to add a label for 400 ppm CO2 in the color bar of this figure.

  – This figure was originally included because in other Arctic regions TA x d18O diagrams very clearly show differences in $pCO_2$ with more freshwater dilution in one direction and more sea ice meltwater dilution in another. We do not find this in our data and that is a

      major difference to what we expected in this region. This will be described in the text but no longer represented as a figure. This figure will be replaced by another figure which more clearly fits our own findings.

        – Figure was replaced

- Figure 7 caption: capitalise M in mixed layer. Lowercase l in lower

        – Will amend this

- 298: Add a point.

        – Will add a point

- Section header 3.3: This section is only descriptive and it appears to me as if would not be complete. I mention this because while the subsection header mentions air-sea CO2 exchange, no flux densities are presented at all (and were notMoreover, I noticed that several references to geographic features / locations which were not discussed before in the manuscript suddenly appear here. I recommend the authors to include these in e.g. Figure 1, should they be relevant for the results/discussion.

        – We will update Figure 1 with the locations mentioned in the text in the revised manuscript. As previously discussed we will not add the flux calculations to the main document though mention will be made of the SeaFlux atmospheric $pCO_2$.

        – Figure 1 has been updated with the locations used in the text

- 209: "respect" instead of "rescpect"

        – Will amend

- 220: I think this sentence is not necessary. IN case the authors would like to use the arguments presented in subsections 4.1 and 4.2 to provide indication of the potential caveats of the approaches used (which would of course be a valid choice), I suggest to shift them to the materials and methods section.

        – We are moving the entire conversation around water mass tracers and our choice to use the polynomial in our normalisation instead of any of the more commonly used normalisation techniques to a supplementary text. This will make the text shorter and more concise and remove the need to add this sentence since this will be moved to the supplement

        – Section moved to supplement

- Figure 10 caption: Since this plot has no units, it is not clear whether the pCO2 values are averages throughout the MLD, or integrated values across the same vertical ranges. Furthermore, in order to convincingly show that pCO2 values within the MLD at all locations are a reliable representation of the flux densities at the the air-sea interface, it would be necessary to see the TS profiles. This because, as pointed out in my comments above, near-surface stratification

can be a non-negligible factor influencing the variability of CO2 air-sea flux densities.

– Units will be added to the plot. We will make the use of the average as opposed to an integrated value clearer in the revised manuscript

– Figure was replaced

- Section 4.1 header: As it stands, this section provides a recollection of factors that might influence TA values in the study area, but misses completely a connection with the data presented above. I kindly invite the authors to revise in order to discuss how the complex setting of the region might have explained their observations.

– We are making modifications in the introduction and the summary of the previous paper that make clear why upstream variability to TA is important to the Northeast Greenland shelf. This should bridge the connectivity issue between the two sections.

– Section was replaced, most of the discussion of TA has been removed

- 231: I am guessing the authors mean to say one out of two major connections to the Atlantic Ocean (?).

– We mean the two regions of export of freshwater to the Atlantic Ocean and will modify the text accordingly

– Where this is first described in the Introduction we make clear that this is one of two major outflow regions together with the Canadian Arctic

- Section 4.2 header: Same comment here as for subsection 4.1. Although the arguments presented here are logical and plausible, the text is (i.e. appears) fully disconnected from the results.

– 4.2 describes the choices of tracer which will be moved to a supplement together with a justification for the choice to use the polynomial normalisation instead of a tracer-based normalisation.

– Everything relating to water mass tracers and normlisation has been moved to the supplement

- 283: I agree with this. However, I have to say the graphical display (and the manuscript text) do not show this as clearly as they could. In particular for figure 6, adding an indication of the 400 ppm CO2 boundary to define sources/sinks would be useful. Along those lines, the authors might want to consider a two color bar to show this more clearly. Along those lines, it should be clearly stated what is the source of this value for atmospheric CO2 at the time of sampling. By looking at the NOAA atmospheric sampling station in Svalbard (78.9067° N, 11.8883° E; https://gml.noaa.gov/dv/iadv/), it seems that 400 ppm could be an overestimate. I therefore invite the authors to re-check.

– The figure will be modified to include a categorical range for $pCO_2$ values which will more clearly show where the atmospheric value is reached. Based on the SeaFlux data product for the geographical region of this study, 400 ppm is an overestimate by about 5-10 ppm for September 2017.

– Figures have been replaced

- 283: Furthermore, and perhaps even more importantly, the fact that the crux of this paper is the variability of the region as a source/sink of atmospheric CO2 and no CO2 flux densities is presented, is a significant drawback.

    – As stated above, the errors associated with the calculation of the $CO_2$ flux in light of the lack of measurements of the required parameters means that actual flux measurements are less useful than simply presenting the measurements and CO2SYS calculated values of the partial pressure due to questionable accuracy. We lack measurements of wind speed at 10 m, of $CO_2$ partial pressure in the atmosphere, and of the parameters required to calculate the gas solubility. This means all of these would be assumed, obtained from reanalysis data, or taken from quite remote measurements (such as from Spitsbergen on the other side of Fram Strait or far further south in East Greenland).

    – We have refocused the paper. We still do not discuss the flux (for reasons previously dfescribed) though we have rephrased to include references to the SeaFlux atmospheric value

- I kindly invite the authors to check for additional data that might be available at the SOCAT database.

    – As stated above, aside from the obvious issues with comparing values from two entirely different measurement techniques, there is data for the month of September only from 2009. This is almost a decade before the data described in this manuscript and the region has changed quite dramatically in the interim. We will make a point of discussing the SOCAT data for the region in more depth to describe the seasonal variability in the region as stated above.

    – Data is not included in main manuscript but has been included in the Supplement

- 291: The authors might want to consider enriching the discussion with additional, recent literature, e.g.: Richaud, B., Fennel, K., Oliver, E. C. J., DeGrandpre, M. D., Bourgeois, T., Hu, X., and Lu, Y.: Underestimation of oceanic carbon uptake in the Arctic Ocean: ice melt as predictor of the sea ice carbon pump, The Cryosphere, 17, 2665–2680, https://doi.org/10.5194/tc-17-2665-2023, 2023.

    – Thanks for mentioning this paper, it is very interesting and a reference to it may find its

way into the revised manuscript.

- 353: In the paper by Sejr et al., it was shown that turbidity correlated well with the pCO2 offset between the two methods. I wonder whether turbidity data during the surveys for this study could be used to "correct" the TA/DIC observations and then compare the obtained pCO2 with SOCAT pCO2 data.
  - Sadly, the CTD did not have a turbidity sensor attached so we cannot use this method and we can't correct the data to compare these properly.

**Changes with respect to comments by reviewer 2**

- While the data and findings certainly deserves publishing, I found presentation lacking and certainly agree with the other reviewer Dr. Arévalo-Martinez: the goal of the study is unclear. Related, and as a consequence of this, there is not a clear underlying methodological strategy, the results are unsystematically presented, and large parts of the discussion does not clearly relate to the results. I think the authors need to make a decision on what problem the study should focus on. I would recommend that the authors undertake a systematic analysis of the factors that drives the spatial pCO2 variability. This is interesting, as the region is little investigated in this regard. This is further elaborated in my Main comment no 1.
  - In light of the complex hydrography of the region a clear determination of the precise drivers is impossible, the sampling density is simply not sufficient. This is a synoptic scale study and many of the most important processes range from micro- to mesoscale (sea ice melt/brine to eddies). What we do is show the region's carbon system does not respond to the generally investigated drivers in the same way as other parts of the Arctic and investigate why that is. We agree that this could be phrased more clearly and will do so in the revised manuscript. Figure a contains plots for each of the four driving variables held constant. We do not feel that such an analysis adds to our understanding of what drives variability on the shelf because there are simply too many different water types and processes to consider. To choose representative values we'd have to choose which water mass would be most representative of the region which with complex hydrography is impossible to do.
  - The entire document has been rewritten to focus on the main factors that in most cases explain the majority of dissovled $CO_2$ variability
- Also, note that some terms are not properly defined, in particular Sp – and sometimes S is used and another time salinity. Figure captions are sometimes insufficient, for example, what the

colors signify. Some instances are mentioned in my specific comments, but this needs to be improved, overall. The English is overall excellent, but many sentences are long and winding, shorter sentences and more use of commas are encouraged.

- – We will update the manuscript to use a single term for salinity and make sure any figure colours are described in the captions.
- – Salinity or S are used throughout. Figures have been replaced. Sentences have been made shorter

- Main comment 1: Drivers of pCO2 variability is presented in section 3.2. As a digression, I find it quite illustrative of some of the issues with the manuscript that the section is entitled "Salinity normalized data" while it is really about drivers of pCO2 variability. That aside, there is a substantial methodological issue here as the section simply presents correlations between pCO2 and more or less related properties. But none of these directly drives pCO2.

  - – This section is not about the drivers of $pCO_2$ variability. It is about the associations/correlations between $pCO_2$ and other variables measured and calculated to determine the possible drivers in the absence of clear patterns in $pCO_2$ w.r.t. The variables T, S, TA, and DIC. We have two water types (EBAW and rAW) that have similar salinities (34.8 and 35 respectively) one of which has been on a round-trip of the Eurasian Basin in the Arctic Ocean and the other has not. We cannot distinguish between the two clearly enough to differentiate their TA. Furthermore, temperatures of freshwater at the surface can be as high as the temperatures of rAW, while the other water types remain close to freezing. TA is not entirely conservative with salinity so single source freshwater dilution inferences cannot be made since these sources all have different slopes and intercepts at 0 salinity and mix in unpredictable ways. Finally, the surface water doesn't sustain the amount of productivity required to dominate exchange of $CO_2$ with the atmosphere due to the surface mixed layer being nitrogen depleted (with the exception of the sea ice proximal stations and regions where direct Atlantic input supplies nitrate to the surface). The region is too highly variable and therefore the sampling density is still too low to point clearly at a single driver or combination of drivers. This was the reason to look at correlation of factors instead of at the main four drivers. We agree that this could be stated more clearly in the text and will add additional descriptions of the water types advected into the region, the superimposed processes, and the seasonal variability in the updated manuscript.

  - – We have refocused the paper on the primary drivers as requested though we are only able to look at these using median-based nonparametric methods

- pCO2 variability is caused by four factors, DIC, TA, and temperature (T) and salinity (S), where the two last ones are thermodynamic drivers affecting the CO2 solubility. The importance of each of these drivers should first be determined, and the authors certainly are in a position to do so as they have DIC and TA data. One way this can be done is to determine pCO2 varying one variable at a time, while keeping the others constant at their mean value, and evaluating the correlations with actual pCO2.
  - In this region the solubility influencing factors (T & S) do not seem to influence the Northeast Greenland shelf in the same way as elsewhere. I agree we should add a section where this is discussed more clearly though as stated above, holding values constant does not really provide the answers. We will amend the revised manuscript to discuss this.
  - We performed a comparison with set values as part of the author comment and this did not really provide any additional answersi and was not included in the manuscript
- Next, the factors driving (in particular) DIC and TA variability should be examined. In particular, whether the variability is caused by freshwater dilution or not – though salinity normalization, and whether the location is dominated by Atlantic waters, meteoric waters or sea ice melt.
  - There is no single source of freshwater to the region. Sources of freshwater into the Arctic Ocean, which is exported through Fram Strait and advected onto the Northeast Greenland shelf through Ekman transport, are extremely variable in terms of TA. They include Pacific inflow freshwater, European Russian and American riverine inputs across different geologies, local Glacial input and direct precipitation. These sources are clearly described in Figure 1a in the manuscript. Ten to eleven percent of global riverine input goes into the Arctic Ocean and the volume of the transported Arctic Ocean freshwater inputs dwarfs the local input sources. As a result, the concept of 'freshwater dilution' in terms of carbon system parameters means something different on the Northeast Greenland shelf than it might elsewhere where there are fewer separate freshwater sources. Since this was unclear from the initial version, we will update the revision with additional background. The hydrography in terms of sea ice melt, meteoric waters or Atlantic waters was the topic of our previous paper. A short summary of those results will be added to the Cruise location/description section.
  - The revised manuscript goes into more detail
- My point here, is that none of the properties shown in Fig 8, directly affects pCO2. Yes, they are related, but not directly. Therefore such a two stage approach is much more preferable: (1) How does DIC, TA, T, and S drive pCO2 variability in the region and (2) what regulates DIC

and TA (mixing/source waters and primary production, where the former can be quantified with the data at hand, while the latter can only be inferred).

- As stated above, the data density is too low to determine the effects of the primary drivers without having to look at individual sub-regions where there would then be too few data to run the analyses. We cannot tell apart water with the same temperature or the same salinity since they may be from different sources and are not conservative in this region.

- Figure was removed

• I think the entire paper could revolve around such an analysis, which would give much clearer goal, methodology and results.

- If we had the data density required to do such an analysis we would have but our current data do not provide a clear enough signal to make this the focus of the paper.

- Manuscript was rewritten

• Main comment 2. I find the description of the water mass fraction analysis at lines 94-114 very inaccessible and confusing, in particular from line 100 and onwards. It is mentioned at the start of this passage that "the lack of knowledge on seasonality and mixing history" is a large source of uncertainty. This obfuscates the real issue, which is the fact that almost all of the data are very close to the Atlantic end-member, and far away from the SIM and meteoric end-member, with salinities close to 30 and above. This problem is in particular acute for the TA-S triangle, which simply cannot separate between SIM and Meteoric water at such high salinities. This shortcoming should be stated.

- The point of this graph was to show that we cannot use these end-members for determining the sea ice melt fraction in this region. We have no way of knowing which is the more correct of the two. Both tracers show the same results for the determination of the freshwater fraction however and can definitely be used there. The points regarding the end member position with respect to freshwater in the S-d18O diagram was extensively discussed in our previous paper. Will add this point to the previous paper summary and move this discussion to a supplement.

- Moved to supplement

• Further, many sentences are very unclear and there is a lack of a real conclusion on what tracers to use. What tracers where used to define the water mass fractions in the end?

- We use water mass fractions only to do the Yamamoto-Kawai 2005 sea ice melt normalisation of TA for purposes of comparison to the polynomial normalisation which is used. For this we used the d18O as per their paper. The discussion on tracers and normalisation of carbon data will be moved to a supplement. The revised manuscript will mention only the

polynomial normalisation. This does not depend on any tracer.

– Moved to supplement

- Finally, the axes used for the panels in the two rightmost columns in Figure 3 (panels b, c, e, f, h, l) do not have the same y and x axis (except panel h), and are not square. This makes it very hard to evaluate deviations from a 1:1 relationship.

– An amended Figure 3 will be moved to a supplement

– Figure replaced

- My recommendations for the water mass analyses is to consider whether to use TA at all or not, to revise the section to improve clarity, to make the conclusions with regard to what tracers were used in the end more clear, and to make sure that each panel in the two rightmost columns have the same x and x axis and is square, so deviations from a 1:1 relationship can be more readily evaluated.

– We don't perform a water mass analysis in this manuscript. The hydrography and water types present on the shelf were extensively discussed in our previous paper. We will mention the hydrography in the summary to be added to the cruise description in the revised manuscript.

– Moved to supplement

- The authors can also consider using Optimum Multiparameter Analysis (https://omp.geomar.de/node2.html) for a more objective water mass analysis.

– This was considered (for our previous paper) and rejected because not enough tracers are available for each source water type with clear, known, and differentiable values to do this properly. Even if they were available, choosing the weights of each parameter can introduce large variability in results.

– Not included

- Main comment 3: I found the discussion section (section 4) quite frustrating to read. The first subsection (4.1) is a review of processes affecting alkalinity in the Arctic – but the pertinence to the presented results is not clear – it does not help us understand what goes on in the study region. Section 4.2 appears completely irrelevant, Section 4.4 summarizes many processes but does not present a real conclusion of what goes on, Sections 4.3 and 4.5 should be merged, and preferentially shortened.

– Since the freshwater in the study region is almost entirely obtained from the Arctic Ocean, these processes are of paramount importance to the analysis of our data. This will be made clearer in the next revision. Section 4.2 is certainly relevant since it shows that water mass tracing in the region has been used in a problematic way (assumed to sum to

0 while this is a pan-Arctic phenomenon and locally plus and minus melt cannot sum to 1 due to advection pathways of the different water masses in the Arctic) but it is indeed not a result of our study and will be moved to a supplement.

– We no longer deal with tracers in the manuscript and their discussion has mvoed to the supplement. The results and discussion have also been merged

**Specific comments:**

As above, since the document has been rewritten, not all comments are applicable to this version of the text

- Line 11, I wouldn't refer to Takahashi et al., 2014 as a modelling effort, it is a data synthesis and interpolation effort.
    - We will amend the text
    - Updated as advised
- Lines 16-17, please include a citation for the later onset of sea ice cover in the region.
    - We will add a reference, e.g. de Steur et. al 2023 show that in the Western Greenland sea the ice
    - We have added a reference
- Line 19, Arctic Amplification is largely a consequence of the sea ice loss: the ultimate cause is global warming. Please rephrase.
    - This is correct. We will rephrase.
    - Arctic amplification is no longer mentioned
- Line 54-57, please provide information on how TS were used to group the profiles – even if this is explained in Wilcox et al 2023, I find that it should be included here as well.
    - We will add a section which summarises the main findings of the previous paper which explains the profile groupings and other major findings pertinent to this manuscript (such as main water source location in the Laptev, low nitrate in the surface waters and the oxygen maximum being 'trapped' in the winter mixed layer) in the cruise description section.
    - A few paragraphs have been added describing the previous cruise
- Line 63-65, I believe the correct unit is µmol kg-1 - micro mol per kg, not mmol kg-1. An uncertainty of 2 mmol kg-1 is 1000 times larger than the usual measurement uncertainty of 2 µmol kg-1
    - Will be amended
    - Has been changed

- Figure 2, please add station positions also to the sea ice cover panels.
    - Will update the figures to include the stations in the sea ice cover panels
    - Figure has been changed
- Line 66, please subscript "2" in pCO2.
    - Will amend
- Line 70-71, the depth of the mixed layer does not have any influence on the direction of air-sea gas exchange, that is only determined by the atmosphere ocean pCO2 difference. The MLD determines the mass of CO2 that can be absorbed/released before equilibrium is reached.
    - Will amend the sentence to better reflect the actual processes
    - These sentences were replaced in the rewrite. Any discussion of the MLD is currently more focused on it's rol in trapping the gases below it as evidenced from our measurement of oxygen
- Section 2.2, a bit of validation of this approach would be worthwhile; I suggest including figures some typical T+S profiles with the thus determined MLD in the supplementary.
    - This would be reproducing our previous paper. Figure 4 in our previous paper clearly shows where in the water column the maximum $N^2$ (Brunt-Väisälä frequency squared) is found with depth for each water type. We will add a sentence to the previous paper summary in the cruise description section to make it clear at which depths the maximum $N^2$ was found for each group and the relationship this has to the $[O_2]$ maximum which is just beneath it, and shows that the area beneath the maximum $N^2$ isn't ventilated. This should be sufficient evidence to support our claim that the $N^2$ maximum can be used to determine the mixed layer depth.
- Line 85, note missing punctuation mark.
    - Will add punctuation mark
- Line 95/Table 1, I find the end member value for alkalinity of 2267.33 at salinity = 34.9 very low. For example, using the TA-S relationship determined for the Nordic Seas by (Nondal et al., 2009) gives a TA of 2304 at S=34.9 (their Eq. 6). Simply inspecting the data available from the Nordic Seas in GLODAP (Lauvset et al., 2021), makes it abundantly clear that 2267 is below what is usually observed at this salinity.
    - This is the average value in our data for that salinity and therefore corresponds most closely to the water found at that salinity in our dataset. It is indeed low compared to Atlantic Water from lower latitudes. The Nondal relationship is arguably not relevant on the Northeast Greenland shelf since the region is not part of the Nordic Seas, rather it is an Arctic outflow shelf region, e.g. it does not receive the majority of its water directly

from (return) Atlantic inflow. The water found in our dataset at a salinity of ∼34.9 is composed of two water types, Eurasian Basin Atlantic Water (which has been transported around the Eurasian Basin in the Arctic Ocean and consequently cooled and modified) - and- return Atlantic Water from the West-Spitsbergen Current. These two water types may have different alkalinities. The latter enters the shelf mainly at depth though with increasing shallow water incursions occurring with decreasing latitude. This will be made clearer in the revised manuscript and figures.

– Low and high values for Atlantic Ocean vs North Atlantic Atlantic Water TA are now referenced explicitly

- Seeing that this very low value was determined based on the author's own measurements I would like the authors to take a second look at this. As described in the 2023 Wilcox paper, the titration system appeared well behaved when analyzing CRMs from Dr. Dickson's lab. However, this does not rule out errors resulting from sample handling. With the fairly low sample volume of 13 mL, the samples might be in particular vulnerable. Further QC of the data and comparisons with other measurements is encouraged. This can be done using point-to-point comparisons, or evaluating how these data fall along a TA-S line compared to other data. In this, keep in mind that that the salinity data from the CTD might also be biased – as apparently no at sea calibration of the CTD sensors was carried out. If the TA data are biased low, the estimated pCO2 will be high. This will have implications for the sink source patterns that are presented; further QC of the data is very important.

– These data were collected on two separate cruises, by a different single individual per cruise and analysed in two separate labs (one for the first and a different one for the second cruise). Although the volume of 13 mL seems low, the laboratory in Manitoba was involved in an Inter-laboratory Comparison of Seawater CO2 Measurements managed by Andrew Dickson's laboratory at Scripps Institution of Oceanography with >80 laboratories and our lab showed very good agreement with certified samples provided by Andrew Dickson's laboratory. Both cruises and labs have samples with low TA. We have looked into our analysis data again and cannot find any errors. We are comfortable with our sample storage and small volume analysis. We have other studies from the area where we find higher TA values using the same procedure but they are from earlier in the season (Rysgaard et al., 2009; Rysgaard et al., 2012; Sejr et al., 2011). As we, to our knowledge, are the only ones to sample for TA and DIC in this area and at this time we will have to rely on our measurements. Finally, where the 'Atlantic Water' value seems low this is because it is primarily 'Eurasian Basin Atlantic Water' which may be different from (return) Atlantic

Water at the same salinity since it has taken a different path and may have been subject to the addition of dissolved $CO_2$ rich brines which may have been buffered by the alkalinity of this layer throughout its transport path around the Eastern Arctic.

– Possible reasons for variable Atlantic Water TA are mentioned in the dataset and all used values are referenced

- Figure 3: Note that "Sp" as used on the axis labels has not been defined.

– We will homogenise the salinity term across the manuscript.

– Salinity or S now used throughout

- Figure 4: Please mention in caption that the color codes refer to the TS groups (Fig 1), here and in all other figures where these color codes are used.

– We will clarify the colours in the figure caption

– The discussion in terms of TS groups was removed from the manuscript

- Line 115: Gas exchange should be included in this list.

– We will add gas exchange to the list. A large part of this section will be moved to a supplement which goes into the choice to use the polynomial to normalise the data instead of more commonly used techniques

– This sentence was lost in the rewrite

- Line 119: Sp does not appear in the equation. Sp and S are both used to refer to salinity in many places in the manuscript, please stick to one of these.

– Will amend

- Line 125, 'the highest salinity for the dominant water mass is usually used as reference'. Is this correct? please include citation/example. My sense is that average salinity is frequently used.

– The reviewer is correct and the section will be updated (Figure to be replaced by Figure b) to reflect this change and also moved to a supplement.

– Changed wording and moved to supplement

- Line 126-127, the sentence here "This makes any …" is very unclear – what is exactly meant?

– What was intended was to argue that the use of reference salinities does not necessarily lead to replicable results in environments that have multiple input sources since the reference salinity may vary. A salinity of 35 is also not representative for any processes taking place in for example the PSW/cold halocline layer since this has already been modified, and a salinity of 32 is not representative of the rAW which may be dominant at slope stations at lower latitudes. Therefore finding the appropriate reference salinity in a multi-layer and stratified environment is somewhat problematic even though it is the only option we

have at this time. This sentence is part of the discussion on water mass tracing and will be moved to a supplement.

– Moved to supplement

- Line 144-150: What exactly is the 'sea ice melt water only correction' (panel c values) and how is this combined with the Friis et al (2003) corrected values to obtain the values shown in panel e. Please provide an exact description of this methodology.

    – The sea ice meltwater correction is the correction as described in Yamamoto-Kawai, et al. (2005) which was referenced previously in the text. Will update the text to reference it here again. The two are combined by running the Yamamoto-Kawai, et al. (2005) sea ice meltwater correction method and then using the result as input into the Friis, et al. (2003) method. A sentence to this end will be added to the manuscript.

    – Modified, clarified, and moved to supplement

- Line 156. Use of the word 'intriguing'. It is not clear from the text that follows, how these differences are 'intriguing', I am anticipating to see some surprises/something not anticipated/not readily explained to be pointed out in the text that follows. This is not the case. Please point out what exactly is intriguing or use a more neutral language.

    – Will remove term 'intriguing'

- Figure 5:

    1. Please note that there is no reference to panel a in the
    2. Please explain what the colors of the points signify.
    3. Please use same range for both x and y axes in panel f.
    4. In panel f, the axis labels should be nTA(predicted) for the x-axis and nTAeSsim for the y axis.
    5. The equations in the legend in panel f has slopes of -282.78 for TA and -329.95 for DIC, this is far the 1:1 relationship, that is stated in the text (Line 149)

        – The new version of this figure (above) will be removed from the main text and places in a supplement

        – Modified and moved to supplement

- Line 150-151, the sentence 'particularly where no d18O data is available in a system....' Does not make sense and needs to be revised.

    – The entire discussion on the choice for polynomial correction will be moved to a supplement with the justification for the choice and referred to from the revised manuscript. This should enhance legibility.

    – Moved to supplement and rewritten

- Line 160, surface layer values – of what?? – are lower.
  - This section will be cleaned up to be more descriptive in the following version of the manuscript and more clearly note when we are speaking of $pCO_2$, TA, or DIC
- Line 161, distinct difference – of what??
  - As above
- Line 162, the broadest range of surface values — of what?
  - As above
- Line 173, how were these points, 1, 2, 3, chosen.
  - Will amend text to reflect these are not 'points' but general regions roughly indicating a triangle
  - Figure removed
- Line 174, frankly, the shape of the d18O:DIC is very similar to that of d18O:TA; one can easily fit most of the points in a triangle.
  - Figure 7 will be replaced by another figure which more clearly makes the point that the off shelf and slope stations where mixing with off shelf water occurs are the locations for high $pCO_2$ and determine the effect.
  - Figure removed
- Line 176-177, while I agree that the most extremely high pCO2 data falls on the line between 1 and 3, they are not clearly associated with the slope 1 group of data, as evaluated from Fig 7a.
  - This graph was included because it sets our region apart from similar studies in the Arctic where on a TA x d18O plot there is a clear pattern when data are coloured by $pCO_2$ showing the effects of sea ice melt and meteoric water dilution independently through changes in slope. Our data show no such thing. This seems to have gotten lost in our manuscript. This will be made clearer in the revision by the replacement of this image with others and by being more specific in the text. Figure removed
- Line 177, the 'by average' here is confusing, the 1-3 mixing line falls at lower TA and higher d18O than the 1-2 mixing line. Please revise.
  - As above
- Figure 6: For pCO2, I would recommend a red-white-blue color scale, with white being the atmospheric levels; that would allow one to more easily discern under- and oversaturated waters.
  - It is unlikely that the journal allows red, white, blue colour scales due to their being unfriendly to some forms of colour blindness but we will investigate more representative

colour schemes and make colour bars categorical for clarity.

- – Red white blue is certainly not an option due to having to tick the 'tested for colourblindness' requirement for submission. But clearer 3 colour schemes have been used where possible

- Figure 10: Please indicate the location of the geographical features discussed in the text (Young Sund, Belgica Trough etc.)

  - – Figure will be updated with labels of any regional features mentioned in the text
  - – Figure 1a in revised manuscript now shows areas mentioned in text

- Section 4.1: These are interesting deliberations, but the relevance and implications for the scientific analyses that are conducted must be made much clearer. How do these processes translate into quantitative uncertainties?

  - – It is not really possible to know what fractions of water comes from each of these regions due to a lack of tracers and the (for example seasonal) variability in measurements of the tracers we do have. This makes quantitative assessment of the alkalinity supply to the Northeast Greenland shelf with so many different source waters and mixing processes impossible at this time.
  - – Reasons why water mass tracing is unlikely to work are briefly touched upon in modified manuscript

- Line, 257, this is a bit of repetition of the text at line 238-241.

  - – We will modify the text to be less repetitive

- Section 4.2. This discussion of the freshwater balance across gateways/in Arctic Ocean regions is not at all relevant to the manuscript, please remove.

  - – Though we certainly consider this discussion relevant to the use of d18O as a tracer for sea ice melt water and the assumption that brine is the negative of melt, this section will be moved into a supplement together with the tracer discussion.
  - – Section removed to supplement

- Line 291-292, please provide a citation for this statement.

  - – Will add a citation e.g. Rysgaard, et al. (2009)

- Line 292-296, this brief methodological review appears irrelevant to the matter at hand.

  - – It is relevant to our discussion that our data do not show the same patterns (clear pattern in of TA x d18O plot when coloured with pCO$_2$ which we do not have) as are seen elsewhere in the Arctic. This should become clearer when the results and discussion section are merged as suggested by the other reviewer.
  - – Results and discussion are merged. Details about water masses and tracers are moved to

supplement.

- Line 296, what/who is 'the same' referring to here?

    – As above

- Line 297, who is 'their' referring to here?

    – If this sentence is reused once the results and discussion are merged, sentence will be updated to include the papers referred to instead of 'their'.

- Line 303-304, 'Sp' and 'salinity' used for the same property, stick to one.

    – This will be changed

    – Changed

---

## Referee Report (RR1)

[revised manuscript text omitted]

**Supplement with CO₂ paper**

AUTHORS

**Contents**

**Goal of this supplement**

The ability of the ocean to dissolve carbon dioxide ($CO_2$) gas is primarily affected by temperature, salinity, the buffer capacity of the ocean (measured as titrated alkalinity) and the amount of total dissolved inorganic carbon (the sum of all inorganic carbon species in solution once released as $CO_2$ gas and measured by coulometric titration). To analyse the carbon chemistry from bottle data they are commonly normalised to remove the effect of salinity (S) (Broecker and Peng 1992; Friis, Körtzinger, and Wallace 2003; Yamamoto-Kawai, Tanaka, and Pivovarov 2005) or temperature (Takahashi et al. 2002, 2009). This allows the analysis of the influence of other processes on the carbon system. Generally, the four main abiotic influences on the carbonate system are temperature, salinity, total alkalinity (TA), and dissolved inorganic carbon (DIC) where the TA is generally considered to be conservative with salinity and the DIC is influenced primarily by autotrophic production and remineralisation (Zeebe and Wolf-Gladrow 2001). When normalising data with respect to salinity in environments where TA is conservative with salinity, analyses can focus on the biology. For surface water transported to higher latitudes from low and mid latitudes, the increase in gas solutbility is is associated with the decrease in temperature (Li and Tsui 1971; Weiss 1970; Millero 2013). For an isochemical water mass, the relationship was established by Takahashi et al. (1993) to be $(\partial \ln pCO_2/\partial T) = 0.0423 \pm 0.0002\ °C^{-1}$ for water taken from the North Atlantic.

The Northeast Greenland shelf is a unique high latitude coastal environment with more possible influences on the carbonate system than in lower latitude open ocean environments. The environment can not be expected to be isochemical, nor is the surface water all cooled. Water found at the surface and originating in the Artic Ocean will be exposed to increasing atmospheric temperatures with decreasing latitude in summer which would reduce the solubility of $CO_2$, while the return Atlantic Water might either heat or cool depending on conditions on the eastern side of Fram Strait, the season during which it arrives on the shelf, and the amount of (melting) sea ice it encounters. Similarly, the other main variables measured to calculate the $CO_2$ have different sources or are subject to complex processes on the shelf.

This supplement is intended to highlight some details which are relevant to but not directly part of the study. The first is a discussion surrounding the use of water mass tracers on the Northeast Greenland shelf and the errors associated with it. The second is a justification for our choice of using a polynomial fit to normalise the data rather than using more common methods. Finally we provide some detail regarding our use of the modified Z-score, and a comparison between our data and that found in the SOCAT and CARINA databases.

**Water mass fractions on the shelf**

In an idealised estuarine environment there is a single freshwater source with which incoming ocean water is diluted. This source can be glacial or riverine, and precipitation is considered either negligible or as part of the same catchment. The TA of the freshwater source can be obtained by performing a linear regression between total alkalinity and salinity and finding the TA at S = 0. In a northern latitude fjord environment dilution of the surface layer by sea ice melt is an additional process. This makes the analysis more complex since sea ice retains TA in the form of the hydrated mineral ikaite ($CaCO_3 \cdot 6\ H_2O$) and so is no longer conservative with the salinity, both in the meltwater influenced layer as well as the underlying water into which the salty but TA-depleted water is mixed. In an idealised fjord with a single meteoric freshwater source and local sea ice formation and melting the sea ice melt influence can be approximated by performing a water mass fraction analysis. This is most frequently done by using a system of linear equations where 2 tracers are used to obtain 3 unknown water mass fractions . The most commonly used tracers are salinity and stable water oxygen isotopic composition (δ¹⁸O), which are independent from one another both for meteoric as well as sea ice freshwater sources, for end-members of Atlantic Water, Meteoric freshwater, and sea ice melt as shown in Equations 1, 2, and 3.

$$f_{sim} + F_{mw} + F_{aw} = 1 \tag{1}$$

$$\delta^{18}O_{sim} + \delta^{18}O_{mw} + \delta^{18}O_{aw} = \delta^{18}O_{obs} \tag{2}$$

$$S_{fsim} + S_{mw} + S_{aw} = S_{obs} \tag{3}$$

where subsripts sim, mw, and aw refer to sea ice melt, meteoric freshwater and Atlantic Water end members and obs to the observed (measured) values.

The Northeast Greenland shelf is not an idealised northern latitude fjord, it is a complex broad Arctic continental shelf which receives multiple advected watermasses and receives additional local inputs. The water advected onto the shelf is not a pure Atlantic Water end member, it is instead comprised of return Atlantic Water, directly from the West Spitsbergen Current and Eurasian Basin sourced Arctic Atlantic Water which is much colder and may have been subject to processes specific to the Arctic that the return current has not including such things as dense water cascades or sedimentary interactions.

The upper water which includes the cold halocline layer and the surface water is influenced by sea ice melt and by the input of 10-11% of global meteoric river discharge (Shiklomanov et al. 2021). Each of the 6 major rivers discharging into the Arctic Ocean has its own average TA and δ¹⁸O values which also vary seasonally (Cooper et al. 2008), Due to these complexities we can't assume that TA is conservative with salinity.

The 3 linear equations & solve for 1 unknown system commonly used to determine the water mass fractions is sensitive to the choice of the salinity and δ¹⁸O for sea ice. Sea ice δ¹⁸O can vary depending on the water from which it was frozen, whether or not it is covered in snow, and on its age (first year versus multiyear ice) (Mellat et al. 2024). For end member values AW (S=35.0, δ¹⁸O=0.3‰), MW (S=0, δ¹⁸O=-20‰) and sea ice melt with S = 2 set to δ¹⁸O of -4, -1, and 0.2‰ respectively entered into the system of linear equations, the lowest negative meteoric meltwater fraction (so an indicator of the size of the introduced error) in our data are -9.9%, -8.2%, -7.8% respectively. It is less sensitive to the salinity of the sea ice. For a δ¹⁸O of 0.3‰ , S = 4 results in a maximum negative freshwater fraction of -7.7% and remains the same (when rounded to 2 significant figures) at S = 0.

[Figure]

Figure 1: Density against temperature with fractions of sea ice melt (top) and meteoric water (bottom). Water mass boundaries (Rudels et al, 2022) in colour and the remnant of the winter mixed layer in the black dashed line. Acronyms UW is Upper Water, PW II is Polar Water 2 which refers to the lower halocline & winter mixed layer in the upstream Nansen Basin. Note that the Atlantic Water sea ice melt fraction is close to 0 while simultaneously, the upper water mixes from high in brine (negative melt) to high in sea ice melt crossing through 0 sea ice melt. Meteoric freshwater (FMW) has negative fractions, primarily at high densities which is clearly in error since meteoric freshwater input can't be negative. It is therefore apparent that the system of linear equations with which the water fractions are calculated is lacking the end-members or end-member values required to properly assign these fractions at each data point, likely due to the high variability of input sources

For representative end-member values of AW (S=35.0, δ¹⁸O=0.3 ‰), MW (S=0, δ¹⁸O=-20 ‰), and for SIM S=2 and the mean δ¹⁸O value of sea ice collected and melted during the second cruise: δ¹⁸O = -2.34 ‰ (Willcox et al. 2023). It can be seen that the Cold Halocline Layer (CHL, from the base of the winter mixed layer at σt=25 to the Polar Water II at σt=27.2) is most influenced by negative sea ice melt (generally interpreted as brine) and all other water, the more dense Polar II and Arctic Atlantic Water as well as the surface water have meltwater fractions of 0 ± 5 %. For the surface water this is not a problem since the meteoric freshwater and Atlantic Water fractions are not below 0. It does pose a problem for the higher density waters (σt > 27.2) where the freshwater and/or Atlantic Water fractions are unrealistically < 0 % (magenta in Figure fig. 1 b) and the sea ice meltwater fraction is lower than those erroneously negative fractions. When Atlantic Water enters the Arctic Ocean, it eventually forms the lower halocline when the warm water is rapidly cooled, by loss of heat to the atmosphere, but also through the melting of sea ice and a meltwater signature in these denser waters could be correct and can not be simply discarded. This issue can't be easily resolved without the use of additional tracers such as the ²³⁶U and ¹²⁹I anthropogenic radionuclides which can differentiate between different Atlantic Waters based on their time spent in transit.

Table 1: End member values used to determine water mass fractions. Meteoric water values for $\delta^{18}$O and TA are those of the Lena river according to (Cooper et al. 2008). Sea ice melt values for $\delta^{18}$O and TA are from own measurements on the shelf

|  | Salinity | $\delta^{18}$O (‰ VSMOW2) |
| --- | --- | --- |
| Sea ice melt | 2 | -2.344 ± 0.746 |
| Meteoric | 0 | -20.5 |
| Atlantic | 35.0 | 0.3 |

**Salinity normalisation of carbonate chemistry**

The TA of return Atlantic Water that has sea ice melted directly into it may be different (say a TA of 2330 diluted with a mean shelf sea ice concentration of ~204 µmol/kg) to the TA of Arctic Atlantic Water that has a similar salinity but may have had brine and meltwater added during multiple years spent in the Arctic Ocean. Simply correcting with the sea ice meltwater fraction therefore may not be sufficient to describe local processes.

The simplest formulation of the salinity normalisation of marine inorganic carbon system data is given by Equation eq. 4 where the reference salinity normalised to is often 35 (Peng et al. 1987). Several modifications to this have been proposed with time including those which involve correc- tions for nutrients (Broecker and Peng 1992).

$$nX = \frac{X_{meas}}{S_{meas}} \cdot S_{ref} \tag{4}$$

where X is the variable to be corrected for, e.g. TA and/or DIC, S is the salinity, and meas and ref
subscripts stand for the field measurements and the reference value respectively.

Whether the resulting normalised data are entirely independent of freshwater flux has been ques-
tioned (Robbins 2001). Later iterations were developed specifically for higher latitudes including
corrections for a TA estimated by linear regression at the point S = 0 (Friis, Körtzinger, and Wallace
2003), and for the calculated sea ice melt fraction (Yamamoto-Kawai, Tanaka, and Pivovarov 2005).
Each of these corrections has associated issues and errors and may not provide useful information,
especially where there are multiple low salinity sources for TA such as shelf environments host to
catchments with differing geology. Although there is an official descriptions of what a reference
salinity is (Wright et al. 2010), it is often either chosen to be 35 or a regionally obtained vari-
able, often the mean salinity. This makes any comparison between different geographical regions
with different dominant water masses and therefore chosen reference salinity for calculated values
subject icomparable. This complexity primarily impacts mixed layer depths (Friis, Körtzinger, and
Wallace 2003) where the meteoric-influenced layer is highest or multiple different sources such as
precipitation, riverine inputs, and sea ice melt, contribute to the dilution. If these normalizations
rely on other assumptions such as those underlying the calculation of sea ice melt fraction from
$\delta^{18}$O, any error in these assumptions will be propagated into any subsequent application using the
normalized data.

The processes controlling the water mass composition and the associated shelf salinity and alka-
linity are complex. In addition, fraction calculations suffer from the ambiguities discussed in the
previous subsection, therefore these data might best be normalized with respect to salinity by the
simple removal of a polynomial-predicted value from the data, rather than attempting to correct for
the assumed representative values for the Northeast Greenland shelf which contains such vastly
variable sources in unknown relative quantities.

For purposes of comparison and to choose the best representative method for the salinity normali-
sation of the carbonate system data, four different salinity corrections were applied (Figure 2). The
first (Figure 2a) is the direct application of the polynomial in Equation 5:

$$X_{pred} = X_{obs} - X_{poly} + X_{meanS} \tag{5}$$

[Figure]

Figure 2: Comparison of normalisation techniques. Application by polynomial fit using the green line with equation TA = -3631.43 + 324.03 S - 4.45 S² (a), traditional salinity normalisation (b), Sea ice correction (c), Meteoric freshwater correction (d), Meterric correction applied to sea ice corrected data (e) and finally a comparison between sea ice + freshwater corrections and the polynomial correction indicating a slope of 1 between them where pred is the salinity-normalised value estimated by the equation, obs is the observational data, poly is the value predicted by the polynomial fit (green line in Figure 2a), and $X_{meanS}$ the mean salinity for the dataset. This method therefore still relies on an arbitrary choice of reference salinity but it reduces the number of assumptions made about external influences on the data such as the calculated fraction of sea ice melt although these have results that are comparable enough to be used interchangeably (Figure 2f).

**Modified Z-scores**

Modified Z-scores rely on the Absolute Median Deviation (MAD) rather than the mean of a dataset and thus allow for the labeling of outliers in datasets where the mean is too sensitive to outliers.

This modified Z-score is calculkated according to Equations 6 and 7.

$$MAD = median_i(|x_i - \tilde{x}|) \tag{6}$$

$$M_i = \frac{0.6745(x_i - \tilde{x})}{MAD} \tag{7}$$

Data can then be flagged as an outlier if $|M_i| > D$. Although Iglewicz and Hoaglin (1993) suggest a D of 3.5, this doesn't adequately flag all outliers in our data. To make sure all outliers based on visual inspection are flagged as such we require D = 1.5.

**Comparison with SOCAT and CARINA data**

Limited Surface Ocean CO2 Atlas (SOCAT) carbon dioxide fugacity measurements and and full depth

CARbon dioxide IN the Atlantic Ocean (CARINA) total alkalinity (TA) and dissolved inorganic carbon (DIC) data are available for the region of this study, however it is both geographically (Figure 1 a.

main text) as well as temporally limited (Figure 4). For the time period (late August and September)

of our study in late fall, there is only SOCAT data available from 2009 and CARINA data from 1994

and 2003 and therefore these data are not ideal for comparative purposes.

[Figure]

Figure 3: Density plots of the modified Z-scores of normalised TA and DIC (a,b) and of the data not flagged as outliers based on different choice of D

[Figure]

Figure 4: SOCAT measured $fCO_2$ (a) and CARINA CO2SYS calculated $fCO_2$ (b) for geographical area on and around the Northeast Greenland shelf compared to data from our study where D = 1.5. The grey dashed line is at 395 µatm, which is representative for the time of our study per Fay et al. (2021)

---

## Author Response (AR2)

**Reviewer Response**

**E. Willcox & co-authors**

**Contents**

**General**

Again we are grateful for the comments of the reviewer. We have made all recommended changes.

Please find our responses below.

E. Willcox and co-authors.

**Comments main manuscript**

- Ln 36. fix spelling benthic-pelagic
    - Done
- Fig 1. "Despite the caption, it would be useful for the reader to see the abbreviations of the water masses either directly on the plot (panel a)), or as part of the figure legend."
    - The water masses are stacked vertically and therefore it is not possible to add abbreviations in a way that wouldn't overcrowd the plot. However the sources/currents and gyre (Arctic + TPD, EGC, Atlantic + RAC, GG) are now in the figure legend.
- Ln 116. Comment replacing word "descriptions": Perhaps starting with "A full description for (…)" would be more appropriate.
    - Done

- Ln 129. revise sentence to avoid duplication: "I would recommend revising here to avoid redundancy ("determining" and "determination")."
    - Replaced with "We estimate the depth of the mixed layer by estimating the pycnocline from the determination of the maximum Brunt-Väisälä frequency"
- Ln 130. include self citation after 'our previous study'
    - Done
- Ln 135. Misspelling of 'meteoric'
    - Done
- Ln 136. Create numbered sections in the supplement and refer to the number in the text of the main manuscript
    - Done
- Figure 3. Change caption to include AOU, add period at end of line.
    - Done
- Ln 170. Add period after 'etc'
    - Done
- Ln 205. Manuscript: "The first part of the sampling period had warmer surface temperatures on the shelf itself, especially in the south along the coast. This is associated with higher fCO2, particularly at higher distances from the Greenland coast and smaller distances to the slope (EGC)." Reviewer: "The plots aimed to substantiate this argument (4a, 4b) do not conclusively show this. In my opinion this would be much better depicted with plots containing maps (in similar style as for Figure 2)."
    - Although we agree that the original statement might be better represented graphically, to give a fair appraisal would likely require the addition of multiple images to show temporal as well as spatial variability and/or almost entirely duplicate Figure 2. This seems like a lot to support a single statement. We have instead amended the text but referred to a new section in the Supplement which includes a map with integrated values with a clear correspnding statement concerning the inadequate temporal representation. We hope this is sufficient.
- Ln 211. Replace 'with a lot of'; to 'considerable'
    - Done

**Comments supplement**

- Title: The sections of this supplement as well as its figures should be numbered such that they can be cited unequivocally in the main document
    - Done
- Fig 1: Plots are presented in horizontal rather than vertical mode. This should be corrected.
    - Changed caption to a, b
- Fig 1: The figure quality is very low. These plots should be replaced.
    - Done
- Ln 115: Misspelled "description"
    - Done
- Fig 2. Misspelled "meteoric"
    - Done
- Fig 3: change "them" to "them."
    - Done
- Fig 3: The a-c) labelling on the individual figures is missing.
    - Added labels, edited caption
- Fig 4: The quality of this figure is too low. I advice the authors to exchange it for a new one with appropriate resolution.
    - Increased quality